# Pertpy: an end-to-end framework for perturbation analysis

Lukas Heumos [1,2,3], Yuge Ji[1,2], Lilly May [1,4], Tessa D. Green [5], Stefan Peidli[6,7,8], Xinyue Zhang[1,4], Xichen Wu[1,4], Johannes Ostner[1,9], Antonia Schumacher [1], Karin Hrovatin [1,2], Michaela Müller [1], Faye Chong[1,10], Gregor Sturm [11], Alejandro Tejada[1], Emma Dann[12], Mingze Dong [13], Gonçalo Pinto[1], Mojtaba Bahrami [1,2], Ilan Gold[1], Sergei Rybakov[1,4], Altana Namsaraeva[1,14], Amir Ali Moinfar [1,4], Zihe Zheng[1], Eljas Roellin [1], Isra Mekki[15], Chris Sander[5], Mohammad Lotfollahi [1,13,16], Herbert B. Schiller[3,17] & Fabian J. Theis [1,2,4] ✉

Advances in single-cell technology have enabled the measurement of cell-resolved molecular states across a variety of cell lines and tissues under a plethora of genetic, chemical, environmental or disease perturbations. Current methods focus on differential comparison or are specific to a particular task in a multi-condition setting with purely statistical perspectives. The quickly growing number, size and complexity of such studies require a scalable analysis framework that takes existing biological context into account. Here we present pertpy, a Python-based modular framework for the analysis of large-scale single-cell perturbation experiments. Pertpy provides access to harmonized perturbation datasets and metadata databases along with numerous fast and user-friendly implementations of both established and novel methods, such as automatic metadata annotation or perturbation distances, to efficiently analyze perturbation data. As part of the scverse ecosystem, pertpy interoperates with existing single-cell analysis libraries and is designed to be easily extended.

Understanding cellular response to stimuli is crucial for describing biological phenomena and mechanisms. Single-cell data have increasingly shifted from observational experiments to perturbation experiments, encompassing genetic modifications, chemical treatments, physical interventions, environmental changes, diseases and combinations thereof. Technologies such as Perturb-seq[1], CROP-seq[2] and Sci-plex[3] leverage single-cell readouts to capture perturbations at scale. By monitoring resulting shifts in intrinsic cell states, single-cell perturbation analyses offer insights into changes in gene programs, shared and divergent responses across tissues, drug targets and interactions, changes in cell type frequency and cell–cell interactions after perturbation.

Statistical and machine-learning-based analysis methods have been developed for these complex data, resulting in the discovery of, for example, cell states associated with autism risk genes[4] or stimulation responses in primary human T cells[5]. However, the size and complexity of high-throughput perturbation screens can pose considerable interpretation challenges, lacking meaningful lower-dimensional representations and additional context regarding cell lines or perturbations. Current perturbation analysis frameworks such as MUSIC[6], ScMAGeCK[7], SCEPTRE[8], GSFA[9] and FR-Perturb[10] primarily focus on CRISPR perturbation analysis, neglecting other perturbation data types and perturbation analysis steps. Furthermore, no current analysis framework exists that scales to genome-scale datasets[11], contextualizes data with public annotations and uses common data structures across tools (Extended Data Table 1). In addition, many tools suffer from maintenance issues or are confined to the R ecosystem, complicating analysis. Other widely used frameworks in the single-cell field, such as scirpy[12] for adaptive immune receptor data and scvi-tools[13] for

probabilistic modeling, have demonstrated the importance of enabling efficient multimodal data analysis while providing flexible building blocks for developers. Inspired by their impact and the lack of efficient frameworks for perturbation data, we present a new framework focused on perturbation data within scverse[14].

Pertpy, a framework for perturbation analysis in Python, is purpose built to organize, analyze and visualize complex perturbation datasets. Pertpy is flexible and can be applied to datasets of different assays, data types, sizes and perturbations, thereby unifying previous data-type-specific or assay-specific single-problem approaches. Designed to integrate external metadata with measured data, it enables unprecedented contextualization of results through swiftly built, experiment-specific pipelines, leading to more robust outcomes. To evaluate methods and obtained representations for perturbations, we implemented a series of shared metrics. The wide array of use cases and different types of growing datasets are addressed by pertpy through its sparse and memory-efficient implementations, which leverage the parallelization and graphics processing unit (GPU) acceleration library JAX[15], thereby making them substantially faster than original implementations (Extended Data Fig. 1). We demonstrate this versatility by applying pertpy to three different, popular, single-cell RNA sequencing (scRNA-seq) perturbation use cases. To show how pertpy can discover new gene programs, we study a CRISPR activation (CRISPRa) screen (Perturb-seq)[16], projecting it onto a meaningful perturbation space and evaluating the effect of different preprocessing strategies. Moreover, we demonstrate how pertpy can be used to deconvolve perturbation responses into viability-dependent and viability-independent components in a large-scale gene expression and drug response screen[17] by integrating metadata from existing databases. Finally, we decipher compositional changes and rank perturbation effects in a triple-negative breast cancer (TNBC) study[18]. Whereas previously, a user would separately download cell line or perturbation information from scattered databases while piecing together analysis tools from different, incompatible ecosystems, it is now possible to efficiently analyze complex perturbation datasets end to end with integrated biological context.

We provide online links to tutorials with more than 15 additional use cases that demonstrate pertpy's usage with datasets spanning a variety of cell lines and perturbation conditions, ranging from CRISPR screens[19] to inflammation[20] and COVID-19 severity states[21]. Pertpy is accessible as an extendable, user-friendly, open-source software package hosted at https://github.com/scverse/pertpy and installable from PyPI. It comes with comprehensive documentation, tutorials and use cases available at https://pertpy.readthedocs.io.

## Results

### Pertpy enables fast and scalable perturbation analyses

Pertpy includes methods for analysis of single and combinatorial perturbations covering diverse types of perturbation data, including genetic knockouts, drug screens and disease states. The framework is designed for flexibility, offering more than 100 composable and interoperable analysis functions organized in modules that further ease downstream interpretation and visualization (Table 1). These modules host fundamental building blocks for implementation and methods that share functionality and can be chained into custom pipelines. To facilitate setting up these pipelines, pertpy guides analysts through a general analysis pipeline (Fig. 1) with the goal of elucidating underlying biological mechanisms by examining how specific interventions alter cellular states and interactions.

The inputs to a typical analysis with pertpy are unimodal scRNA-seq or multimodal perturbation readouts stored in AnnData[22] or MuData[23] objects. Although pertpy is primarily designed to explore perturbations such as genetic modifications, drug treatments, exposure to pathogens and other environmental conditions, its utility extends to various other perturbation settings, including diverse disease states where experimental perturbations have not been applied.

**Table 1 | Summary of implemented methods**

| Analysis step | Tool or algorithm | Original authors |
|---|---|---|
| Datasets | Data loaders | Peidli et al.[43] |
| Metadata annotation | API requests to public databases | Novel |
| gRNA assignment | Threshold-based Poisson–Gaussian mixture model | Adamson et al.[66] Repogle et al.[11] |
| Differential gene expression | 'Formulaic' interface | Novel |
| Pooled CRISPR screens | Mixscape | Papalexi et al.[19] |
| Differential abundance | Milo scCODA 2.0 tascCODA 2.0 | Dann et al.[39] Büttner et al.[37] Ostner et al.[38] |
| MCPs | DIALOGUE | Jerby-Arnon and Regev[40] |
| Enrichment | Drug2Cell | Kanemaru et al.[67] |
| Perturbation response evaluation | Distances and metrics Augur CINEMA-OT | Novel Skinnider et al.[68] Squair et al.[69] Dong et al.[41] |
| Embedding | Perturbation spaces | Novel |

The first data transformation step assigns guide RNAs (gRNAs) to cells. These gRNAs are short RNA sequences that direct Cas9 nuclease to specific genomic targets. In single-cell CRISPR screens, each cell typically receives one gRNA (low multiplicity of infection (MOI)), although some experimental designs allow for multiple guides per cell (high MOI). This makes accurate guide-to-cell assignment crucial for linking phenotypic changes to specific genetic modifications. Pertpy provides a thresholding and a Poisson–Gaussian mixture model[11] approach that has been shown to perform well in recent benchmarks[24], accommodating both low and high MOI scenarios. This assignment step is required for downstream analyses, including quality control metrics, perturbation efficiency assessment and statistical aggregation of phenotypic effects across cells containing identical guides.

In a second step, confounding factors such as unwanted technical variation and other single-cell-specific quality control issues are addressed. Technical variation between experimental batches, arising from differences in sample processing, reagent lots or sequencing runs, can introduce systematic biases that confound biological signals. These so-called batch effects are particularly challenging in perturbation experiments where treatments may be applied across multiple experimental rounds or where controls are processed separately from perturbed samples. Complexity is further compounded when studying combinatorial perturbations, where systematic batch variations could be mistaken for interaction effects between different treatments. As pertpy is integrated with the scverse ecosystem, users of pertpy can seamlessly integrate established batch correction methods[25,26] to disentangle technical artifacts from true perturbation responses.

After diligent quality control, a typical analysis with pertpy starts by curating the perturbation annotations against ontologies such as Cell Line Ontology[27] or Drug Ontology[28] and enriching the perturbations with additional metadata obtained from Cancer Dependency Map (DepMap) and Genomics of Drug Sensitivity in Cancer (GDSC)[29] for cell lines, Connectivity Map (CMap)[30] for mechanisms of action and the PubChem[31] and ChEMBL[32] databases for drugs (Methods).

The application of CRISPR can exhibit variable efficacy in affecting gene expression. Pertpy's fast Mixscape[19] implementation accounts for this by classifying targeted cells based on their response to a perturbation, analyzing each cell's perturbation signature to determine if the cell was successfully perturbed (Methods and Extended Data Fig. 1). As the number of applied perturbations increases, comparing and

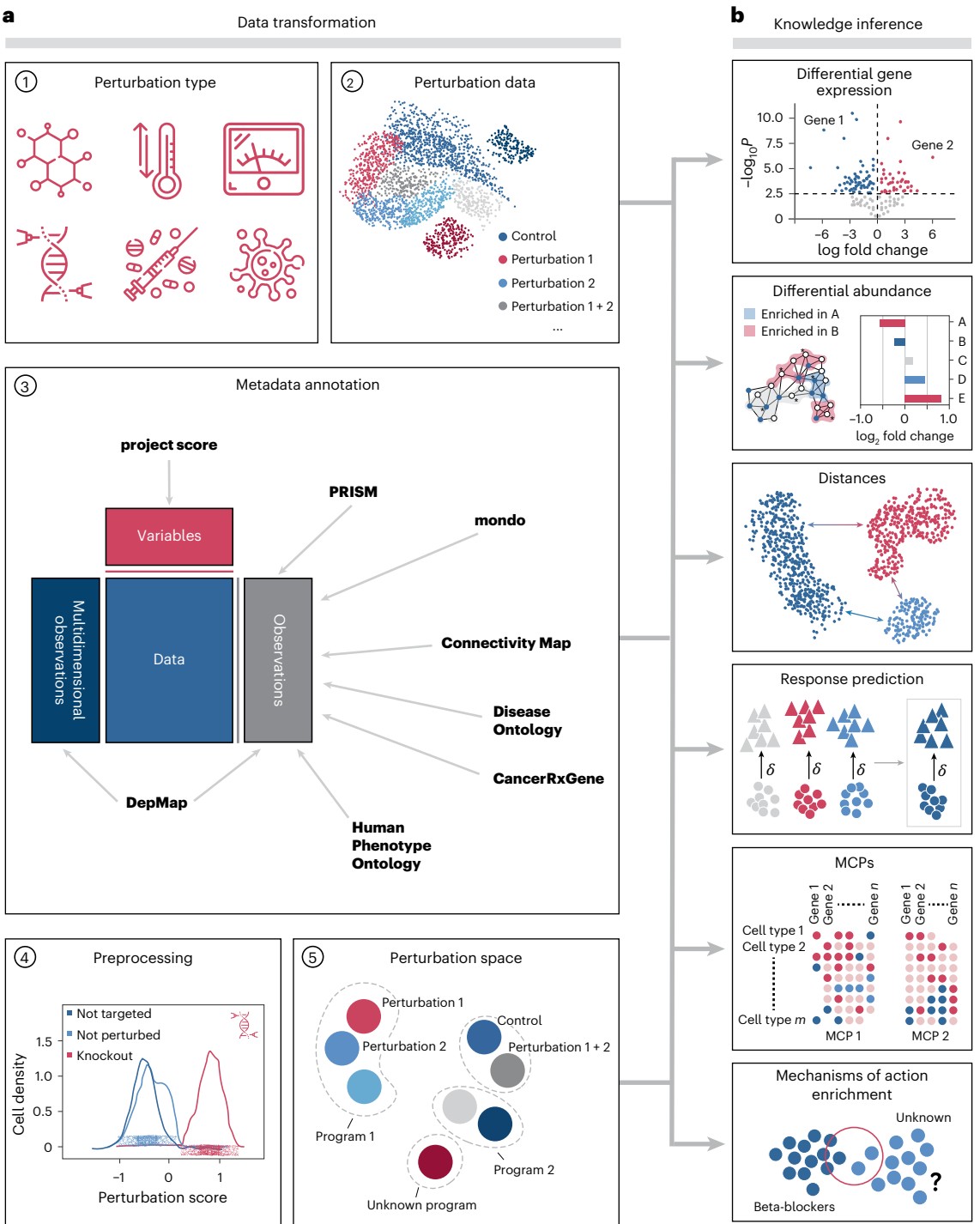

**Fig. 1 | Modules of the pertpy framework. a**, Unimodal or multimodal single-cell perturbation data originating from genetic modifications, chemical treatments, physical interventions, environmental changes or diseases are enriched with metadata from several databases. During preprocessing, confounding factors such as cell cycle and batch effects may be removed. Targeted cells are labeled as successfully or not successfully perturbed. Together, these modules enable the calculation of a meaningful perturbation space. **b**, Pertpy enables downstream analyses, depending on the question of interest. These include differential expression analysis, response prediction, determination of MCPs, calculation of distance between perturbations and mechanism of action enrichment.

interpreting them becomes increasingly challenging. Pertpy provides several distinct ways to learn biologically interpretable perturbation spaces that depart from the individualistic perspective of cells, instead generating a single embedding per perturbation that summarizes cellular responses (Methods). This specialized space, termed a perturbation space, represents the collective impact of perturbations on cells and serves as potential input for downstream analysis[16,33]. Generally, pertpy's analysis pipeline can be adapted depending on

whether the experiment involved multiple cell types or a number of experimental perturbations.

Gene expression changes between experimental conditions are crucial for understanding cellular responses to perturbations. Differential gene expression analysis helps researchers identify which genes significantly change their expression levels when cells are exposed to different stimuli or treatments. Although scanpy[34] is widely used for single-cell analysis, it lacks support for complex experimental designs that account

for multiple conditions, batch effects and nested comparisons simultaneously. Pertpy fills this gap by providing an intuitive interface for differential gene expression that supports complex designs and contrasts, which is needed for multi-condition data (Methods). Currently, pertpy supports PyDESeq2[35], edgeR[36], Wilcoxon tests and *t*-tests. This interface is accompanied by a suite of plotting functions including visualizations such as volcano plots, paired sample expression plots and multi-condition heatmaps. Going beyond differential gene expression at scale, both annotated metadata and differentially expressed genes can be used as input for further pertpy modules such as gene set enrichment tests to uncover the biological effects induced by the perturbations (Methods).

Tracking cell type compositional shifts is crucial for understanding the underlying mechanisms of disease progression, tissue regeneration and developmental biology, offering insights into cellular responses and adaptations. Pertpy offers two distinct methods for detecting compositional shifts, both utilizing a common MuData-based data structure. If labeled groups are available, pertpy provides accelerated and scalable implementations of scCODA[37] 2.0 and its cell type hierarchy-aware extension tascCODA[38] 2.0 (Methods and Extended Data Fig. 1). Both approaches employ Bayesian methods to elucidate cell type compositional changes. If no labeled groups are available or continuous proportions are expected, such as during developmental processes, pertpy implements a scalable version of Milo, previously unique to the R ecosystem[39], which conducts differential abundance tests by assigning cells to overlapping neighborhoods within a *k*-nearest neighbor graph (Methods).

Understanding how cells function together within tissues is a major challenge. Multicellular programs (MCPs) refer to the orchestrated activities of various cell types that collaborate to create complex functional structures at the tissue scale. Pertpy's fast implementation of DIALOGUE[40] uncovers MCPs through a combination of factor analysis and hierarchical modeling, owing to a fast input-order-invariant linear programming solver and a new, fast test to determine significantly associated MCP genes (Methods).

Not all cell types are equally affected by perturbations. Pertpy's fast implementation of Augur (Extended Data Fig. 1) ranks cell types based on their response to perturbations by training machine learning models to predict experimental labels within each cell type and then ranking these cell types by the models' accuracy metrics across multiple cross-validation runs (Methods). Furthermore, understanding the dynamics of cellular response to various stimuli is crucial when experimental exploration of all possible conditions is unfeasible. CINEMA-OT[41], via scalable pertpy implementation, extends this concept by distinguishing between confounding variations and the effect of perturbations, achieving an optimal transport match that mirrors counterfactual cell pairings (Methods). These pairings enable analysis of potentially causal perturbation responses, allowing for individual treatment effect analysis, clustering of responses, attribution analysis and examination of synergistic effects.

For accurate statistical comparison and measurement of perturbation effects, it is essential to employ distance metrics between cell groups. A suitable metric quantifies divergence or similarity in expression patterns of cells under different perturbations, enabling inference of unique or common mechanisms. Different types of distance metrics make varying assumptions on the shape of the data and emphasize specific aspects of difference. For instance, optimal transport-based distances, such as the Wasserstein distance[42], assume correspondence between cell populations, whereas the Mahalanobis distance focuses on covariance structures and scale differences within the data. To capture a wide range of distance metric types, pertpy implements more than 18 different metrics, including, but not limited to, the Euclidean distance (E-distance)[11,43] and the Wasserstein distance (Methods). All included metrics can also be used for perturbation testing through Monte Carlo permutation testing, allowing for the statistical evaluation of perturbation distinguishability and efficacy (Methods).

Built on the scverse[14] ecosystem, pertpy ensures seamless interoperability with existing single-cell omics workflows and can be combined with tools such as decoupler-py[44] and NetworkCommons[45] for tasks such as context-specific inference of protein interaction networks while being purposefully extensible to address new challenges. Base classes for additional perturbation spaces, distances, differential gene expression tests and other components are provided to facilitate swift development. We additionally provide a dataset module with more than 30 public loadable perturbational single-cell datasets in AnnData and MuData format, building upon and extending scPerturb[43] to kickstart analysis, development and benchmarking with pertpy. The metadata of the datasets were curated against public ontologies to enable swift dataset integration and large-scale machine learning, including foundational models.

## Learning and exploring perturbation representations with pertpy

To demonstrate pertpy's ability to learn meaningful perturbation spaces, we examined a publicly available CRISPRa screen dataset initially presented by Norman et al.[16], consisting of 111,255 single-cell transcriptomes of K562 cells subjected to 287 single gene and gene pair perturbations (Fig. 2a). We use this dataset to show how genetic interactions through combinatorial expression of genes lead to cellular and organismal gene programs and phenotypes. We further use pertpy to investigate how different perturbation-specific preprocessing strategies affect the outcome. In particular, we examine whether different strategies may inadvertently remove true biological signals, such as the cell cycle effects induced by *CDKN2A* perturbations.

After initial preprocessing (Methods), we test three perturbation-specific processing strategies: (1) computing cell-specific perturbation signatures based on the 20 nearest neighbor control cells of a perturbed cell and filtering out targeted cells that escaped perturbation based on this signature (Methods); (2) computing cell-specific perturbation signatures using all control cells within the same Gel Bead-in-Emulsion (GEM) group (that is, cells processed in the same sequencing lane) to detect and filter out unperturbed cells (Methods); and (3) no perturbation-signature-based filtering of cells.

Pertpy's Mixscape[19] implementation supports strategies (1) and (2), facilitating comparison of preprocessing strategies. After applying each of the three strategies, we project the normalized gene expression of the remaining cells into a perturbation space using the penultimate layer of our multilayer perceptron (MLP)-based discriminator classifier for each processing strategy (Methods and Extended Data Fig. 2). We found that all strategies yielded similar perturbation spaces (Extended Data Fig. 2i), suggesting that, for this dataset, the approach without perturbation-signature-based cell filtering is preferable. This is expected because the CRISPRa approach used for this dataset does not suffer from cells escaping a perturbation through in-frame mutations, as would be expected in CRISPR–Cas9 screens.

Examining this perturbation space, we observe that explicitly training the classifier to distinguish between individual perturbations results in clustering of perturbations with similar effects on the cell, as indicated by the affected gene program as originally labeled by Norman et al.[16]. We assessed the importance of individual input genes in the classifier's assignment of a cell to a specific perturbation using integrated gradients[46] (Methods). By averaging these feature importances for each annotated gene program, we demonstrate that the classifier prioritizes the respective targeted genes from the set of 4,000 highly variable input genes (for example, *KLF1* for the pro-growth program), highlighting their relevance to the prediction (Extended Data Fig. 3a). In addition to validating known annotations, evaluating data in perturbation space also allows for refinement of previous annotations. For instance, the perturbation *TP73*, characterized as a pioneer factor gene program in the original publication[16], clusters with the G1 cell cycle perturbations when embedded using the

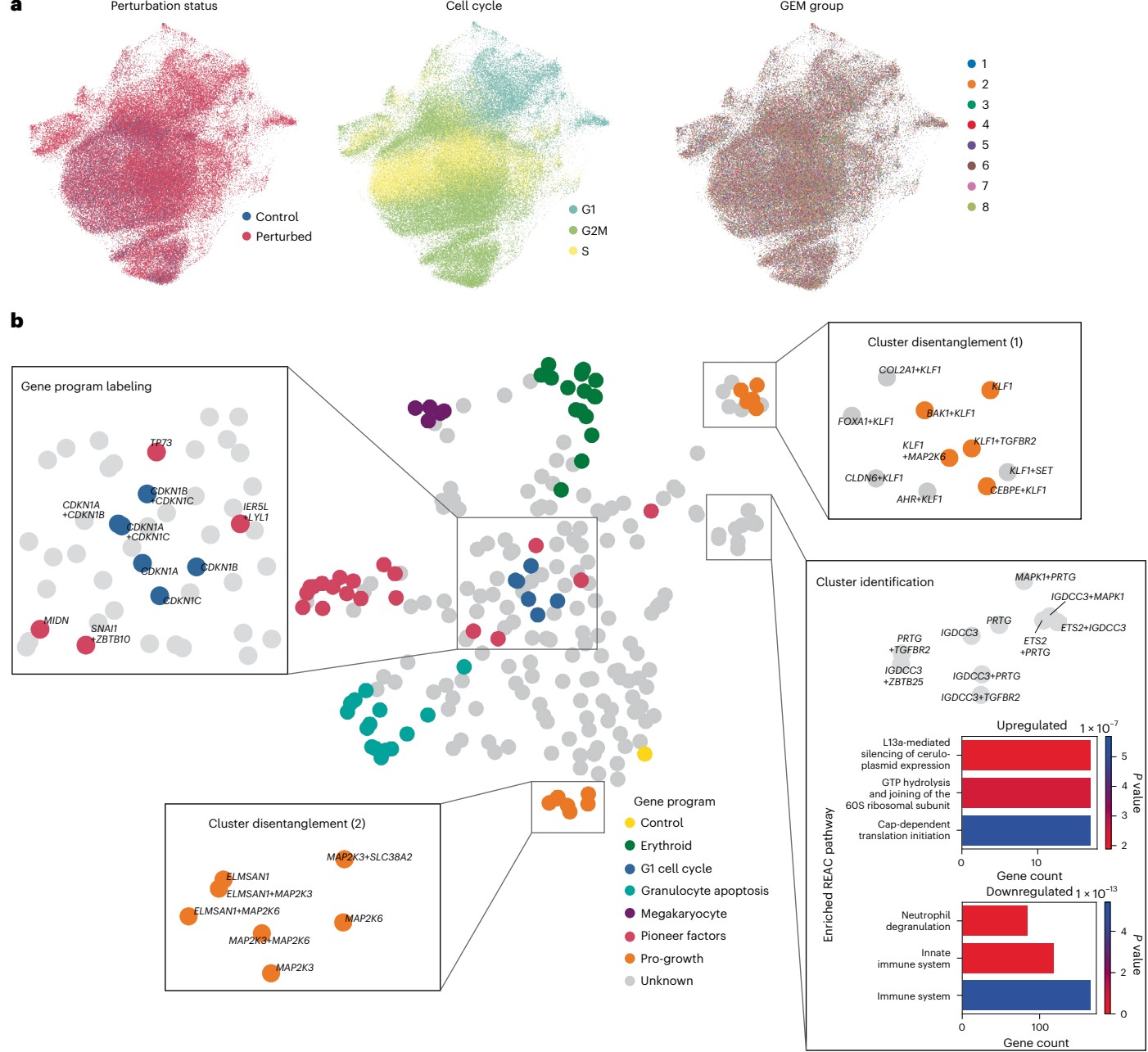

**Fig. 2 | Learning a unified perturbation space in combinatorial CRISPRa perturbation scRNA-seq data via pertpy's perturbation space pipeline.** **a**, UMAP representation of the preprocessed dataset, colored by perturbation status, cell cycle phase and GEM group (that is, batch of cells processed in the same lane on a 10x Genomics chip). **b**, Perturbation space highlighting gene programs that were originally labeled by Norman et al.[16] and details of specific subclusters of interest.

discriminator classifier. This can be explained by the profound influence of TP73 on the cell cycle[47]. Moreover, what the original authors identified and labeled as a single pro-growth gene program cluster can now be differentiated into two distinct clusters (mean squared error (MSE) distance between the two subclusters: 0.46; mean pairwise MSE distance between all gene programs: 0.29; Extended Data Fig. 3b). Indeed, we found that although both clusters comprise perturbations targeting genes important for cell growth, one cluster mainly targets genes encoding Krüppel-like factors (KLFs), whereas the other comprises perturbations of mitogen-activated protein kinase (MAPK) encoding genes. Projection of data into the perturbation space also allows for an in-depth exploration of clusters without gene program annotation, enabling identification of a previously unannotated cluster comprising perturbations with a downregulating

effect on the neutrophil degranulation pathway (Fig. 2b). This use case demonstrates the simplicity and effectiveness of combining several of pertpy's modules into a new analysis pipeline, spanning from quality control over perturbation space to the annotation of previously unlabeled gene programs.

## Pertpy streamlines discovery for complex perturbation experiments

Advancements in multiplexing technologies have markedly increased the number of cell states that can be profiled in one experiment, resulting in large perturbation screens. McFarland et al.[17] introduced MIX-Seq, an experimental assay that enables multiplexing of different cell lines within a single sequencing run. We use pertpy to efficiently analyze a dataset comprising 172 cell lines and 13 drug treatments[17].

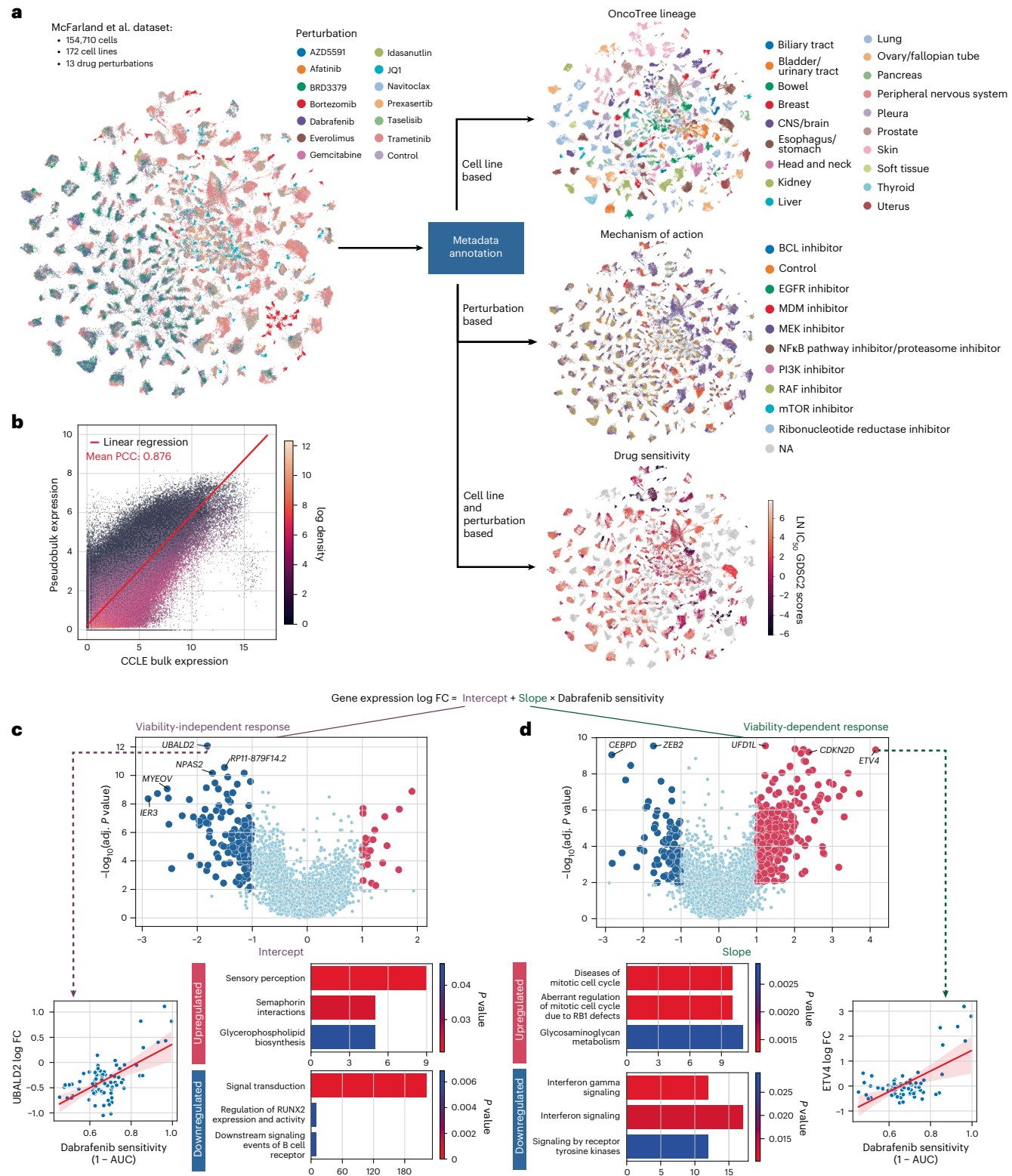

**Fig. 3 | Deconvolution of viability-related response signatures in scRNA-seq drug screen data. a**, Overview of the chemical perturbation dataset. Cell lines and perturbations were annotated with pertpy with additional metadata facilitating detailed analysis. **b**, Linear regression model between single-cell expression data and GDSC profiles shows high correlation, reinforcing the high quality of the dataset. **c**, Volcano plot showing the value and significance (two-sided $t$-test, Benjamini–Hochberg corrected) of the intercept of the fit linear regression models for each gene (top), indicating the viability-independent response. An example linear regression (±95% confidence interval) for the gene

*UBALD2* (bottom left) shows that a change in *UBALD2* expression in a cell line is observable, irrespective of the respective cell line's sensitivity to dabrafenib treatment. The top genes were used to perform GSEA (bottom right), with enrichment $P$ values computed using blitzGSEA[65], which applies Kolmogorov–Smirnov tests and gamma distribution fitting. The figure design is inspired by Fig. 2c in the original publication that introduces the dataset[17]. **d**, The same as in **c** but for the slope of the linear regression models, indicating the viability-dependent response. adj., adjusted; CNS, central nervous system; FC, fold change; PCC, Pearson correlation coefficient; NA, not available.

Pertpy reduces annotation and quality control to just a few steps. Its metadata module annotates cell lines with tissue-of-origin, cancer type and bulk expression profiles from the disease ontology OncoTree[48] and the Cancer Cell Line Encyclopedia[49] (CCLE). Compounds are annotated with their targets and mechanism of action from DepMap[50], GDSC[29] and CMap[30] (Methods). After annotation, pertpy enables immediate visualization for exploratory analysis (Fig. 3a). Additionally, annotated bulk expression allows users to compare RNA profiles of their cell lines with established public datasets, providing rapid quality control functionality. Comparative analysis of the dataset revealed an average Pearson's correlation coefficient of 0.88 across all cell lines (Fig. 3b), demonstrating substantial consistency with the cell line passages cataloged in the DepMap CCLE database and enabling the integration of additional screening data from the DepMap PRISM project[51].

Pertpy significantly streamlines the replication and extension of the original analyses by McFarland et al.[17]. We used pertpy to fetch and annotate area under the dose–response curve (AUC) values for each cell line and perturbation pair from GDSC and PRISM (Methods). This allows us to easily replicate the original statistical method to uncover viability-dependent and viability-independent gene expression associations. We selected a different drug from the original analysis[17], the BRAF inhibitor dabrafenib[52], and used pertpy to compute post-treatment log fold changes across 95 cell lines (Methods). We interpret the intercept and slope of the linear regression on dabrafenib sensitivity ($1 - $ AUC) to be the viability-independent and viability-dependent responses of the respective gene to dabrafenib (Methods and Fig. 3c,d). Notably, we found that cancer-progression-linked genes *ETV4*, *CDKN2D* and *MYEOV*[53] displayed significant variation in their fitted response parameters (Fig. 3c,d). Additionally, our analysis identified enrichment of genes involved in interferon signaling in viability-dependent genes, consistent with initiation of an immune-mediated cell death response to dabrafenib (Fig. 3d). Interestingly, protein translation pathway genes were upregulated in the viability-independent effects of dabrafenib, a response previously noted with dabrafenib[54] but with no mechanistic information until now. This mechanism is distinct from dabrafenib's putative mechanism of action, *BRAF* inhibition, which targets an orthogonal cell survival pathway. Pertpy's ability to efficiently manage, analyze and supplement complex experimental design with additional datasets underscores its utility in conducting sophisticated biology-informed analyses. This streamlined approach greatly enhances the depth of biological insights discoverable.

### Pertpy enables deciphering effects of perturbations on cellular systems

Understanding the complex interplay between the immune system and the tumor microenvironment (TME) is crucial for unraveling cancer progression. This is particularly important in solid tumor entities, such as TNBC, a rare, aggressive breast cancer subtype that lacks estrogen, progesterone and human epidermal receptors, rendering it unresponsive to standard receptor-targeted therapies[55]. Single-cell transcriptomics of breast cancer tumors has uncovered distinct T cell subtypes and the involvement of plasmacytoid dendritic cells in promoting immunosuppression within the TME in TNBC through tumor–immune crosstalk[56], which is a significant driver of treatment resistance[57]. Studies have further elucidated TNBC-specific features and differential responses to neoadjuvant chemotherapy (NACT) and immunotherapy, highlighting the role of programmed cell death protein 1 (PD-1) and programmed cell death ligand 1 (PD-L1) pathways in modulating treatment outcomes[58]. Therefore, we set out to demonstrate how pertpy can be used to investigate treatment responses using a publicly available dataset of 22 patients with TNBC treated with NACT with and without additional PD-L1 inhibitor paclitaxel[18], initially presented by Zhang et al.[18] (Methods and Fig. 4a,b).

To rank perturbation effects, we used pertpy to calculate the MSE distance between pre-treatment and post-treatment patients of the four groups, selected for its strong performance on independent benchmarks[59]. We found that patients responding to NACT alone had a greater distance between pre-treatment and post-treatment expression profiles compared to responders to anti-PD-L1 and NACT combination therapy, implying that the latter led to potentially a less intense response or was used in cases with a worse prognosis.

To identify cell types involved in treatment response, we investigated shifts in cell type composition induced by the treatment. Tracking cell type shifts is essential for understanding disease progression, tissue regeneration and treatment responses, revealing key insights into cellular adaptations. We applied pertpy's implementation of the Bayesian model scCODA[37] 2.0 to the dataset per treatment (Methods). We found compositional shifts for NACT treatment in CD4 central memory, CD8 effector memory, CD8 tissue-resident memory and naive T cells between disease stages but not for combination therapy (Fig. 4d). To better understand whether cell types that are subject to compositional shifts are a part of a common cell circuit, we set out to find shared gene expression signatures in several cell types that jointly act as tissue-level units, so-called MCPs[40].

We applied pertpy's implementation of DIALOGUE[40], which finds MCPs using matrix decomposition in conjunction with a novel, fast input-order-invariant linear programming solver, to the TNBC treatment dataset, calculating 10 MCPs that can be assessed for association with treatment response (Methods). Exploratory analysis of average MCP2 scores across seven distinct cell types in each patient (Extended Data Table 2) indicated a potential association with treatment response for both treatment groups, based on cell-type-specific *t*-tests (adjusted $P \le 1.1 \times 10^{-1}$) (Extended Data Figs. 3a,b and 4a,b). Initial investigations of the MCP2-associated genes suggest involvement in heat shock protein activity and cytokine signaling (Methods, Extended Data Fig. 4 and extended data materials), including an interaction between interleukin 7 (IL-7) and its receptor IL-7R in T cells, which are known to have an antitumor role across diverse cancers[60]. Increased IL-7 activity may contribute to suboptimal treatment outcomes by affecting T cell behavior and elevating levels of MCP2-associated genes *JUN*, *FOS* and *FOSB* (Extended Data Table 3 and Extended Data Fig. 5), which are key components of the AP-1 complex that can either inhibit or promote tumor growth, depending on the context[61].

## Discussion

Pertpy facilitates the end-to-end analysis of complex perturbation datasets with a versatile toolbox of interoperable components, encompassing metadata annotation, data analysis and visualization tools. Through shared infrastructure and modules and with collaboration with original authors, we developed improved versions of widely used methods that were originally unmaintained or easily available only to the R community, making them widely available to the Python community as well. Our community effort will ensure that all of these methods are jointly maintained and further developed. We demonstrated pertpy's flexibility through several use cases, including the identification of perturbation-specific gene programs using a CRISPRa screen (Perturb-seq) dataset, deconvolution of viability-related response signatures in a chemical perturbation dataset and deciphering treatment response to drugs in TNBC. Many further use cases can be found in pertpy's extensive online tutorials.

As perturbation datasets grow larger and incorporate additional modalities such as spatial transcriptomics, we anticipate the development of specialized methods for analyzing multimodal perturbation data. By combining efforts such as Squidpy[62] and pertpy, additional functionality designed for spatial perturbations to uncover, for example, differentially regulated neighborhoods, could be made widely available. To scale to datasets with hundreds of millions of cells, such as the recently published Tahoe-100M[63] dataset, further optimizations in

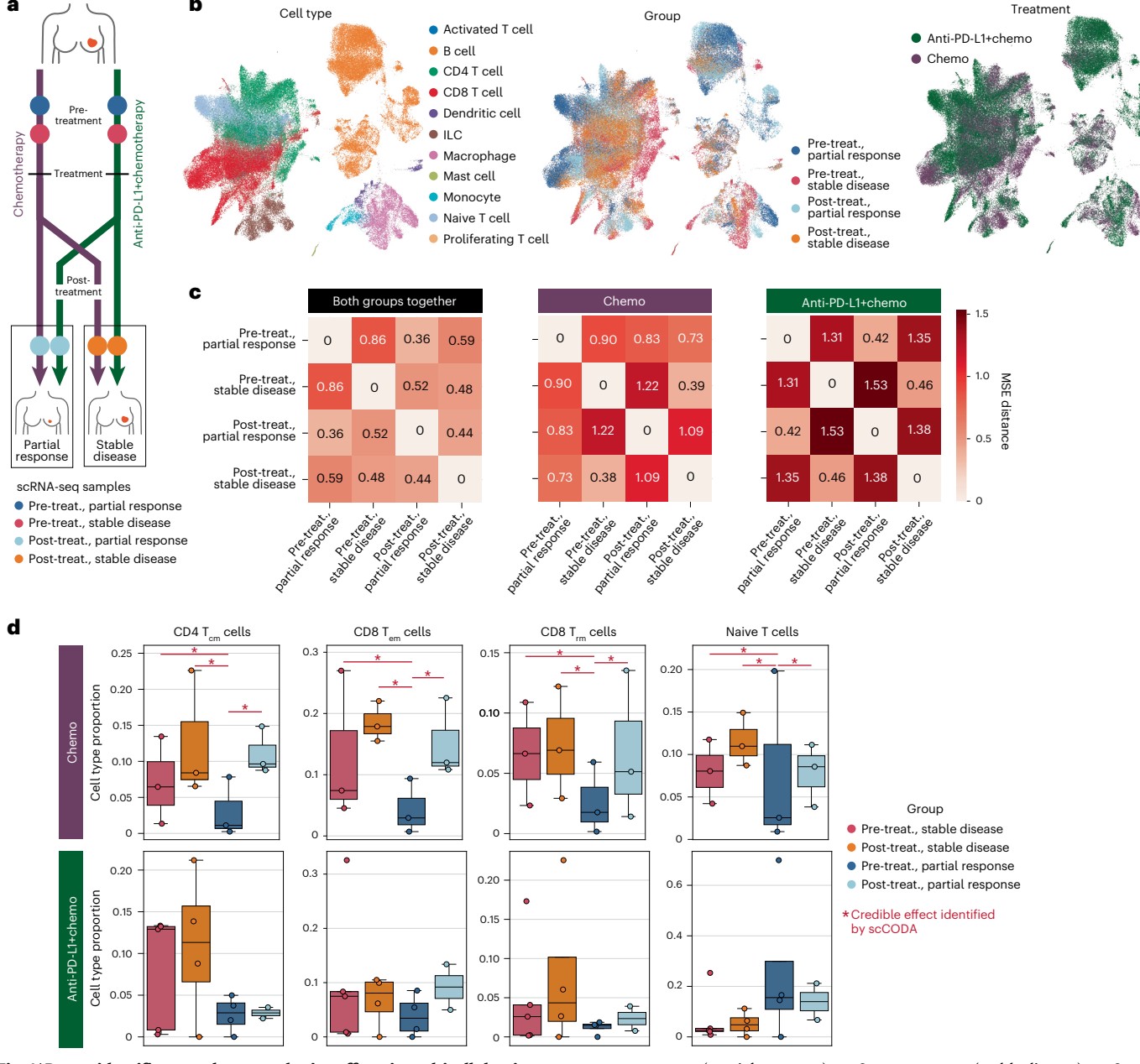

**Fig. 4 | Pertpy identifies complex perturbation effects in multicellular tissue as demonstrated on a TNBC treatment dataset. a**, Schematic overview of the experimental design. **b**, scRNA-seq of tissue from 15 patients with TNBC, comparing pre-treatment and post-treatment responses to anti-PD-L1 therapy and NACT. **c**, MSE distance between treatment responses shows higher distances between partial responses and stable disease. **d**, scCODA analysis shows significant compositional changes for patients treated with chemotherapy. For the chemotherapy cohort, the number of biological replicates was $n = 3$

pre-treatment (partial response), $n = 3$ pre-treatment (stable disease), $n = 3$ post-treatment (partial response) and $n = 3$ post-treatment (stable disease). For the anti-PD-L1 cohort, the corresponding numbers were $n = 4$ pre-treatment (partial response), $n = 5$ pre-treatment (stable disease), $n = 2$ post-treatment (partial response) and $n = 4$ post-treatment (stable disease). Box plots indicate the median and quartiles. ILC, innate lymphoid cell; $T_{cm}$, central memory T; $T_{em}$, effector memory T; $T_{rm}$, tissue-resident memory T; treat., treatment.

---

pertpy through out-of-memory implementations using Dask are necessary, following the approach pioneered by recent Scanpy improvements.

Finally, we expect pertpy to support the creation of perturbation atlases through harmonized data collection, the generation of meaningful perturbation spaces and the evaluation of these spaces using pertpy's distance metrics. Such atlases can comprehensively characterize cell types under various conditions to capture the wide array of inducible cell states beyond their basal states. Enabled by perturbation dataset collections such as scperturb[43] (available in pertpy) and PerturBase[64] (extends scperturb with more recent datasets), we expect such atlases to become essential for the development of robust

and generative foundation models where perturbation analysis is a key task that can be confidently evaluated with pertpy's metrics.

We expect pertpy to lead to more robust biological discoveries through its capability of enriching measurements with biological metadata. As an extendable and interoperable framework, we anticipate that pertpy will enable future robust perturbation analysis methods, tackling the growing complexity and multimodality of perturbation data.

## Online content

Any methods, additional references, Nature Portfolio reporting summaries, source data, extended data, supplementary information,

acknowledgements, peer review information; details of author contributions and competing interests; and statements of data and code availability are available at https://doi.org/10.1038/s41592-025-02909-7.

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

[1]Institute of Computational Biology, Helmholtz Center Munich, Munich, Germany. [2]TUM School of Life Sciences Weihenstephan, Technical University of Munich, Munich, Germany. [3]Research Unit Precision Regenerative Medicine (PRM), Comprehensive Pneumology Center (CPC); Member of the German Center for Lung Research (DZL), Munich, Germany. [4]School of Computation, Information and Technology, Technical University of Munich, Munich, Germany. [5]Harvard Medical School, Ludwig Center at Harvard, DF/HCC Cancer Center, Broad Institute, Boston, MA, USA. [6]European Molecular Biology Laboratory, Heidelberg, Germany. [7]Charité – Universitätsmedizin Berlin, Corporate Member of Freie Universität Berlin and Humboldt-Universität zu Berlin, Institute of Pathology, Berlin, Germany. [8]Institute for Biology, Humboldt-Universität zu Berlin, Berlin, Germany. [9]Department of Statistics, LMU Munich, Munich, Germany. [10]Helmholtz Pioneer Campus, Munich, Germany. [11]Boehringer Ingelheim International Pharma GmbH & Co. KG, Biberach, Germany. [12]Wellcome Sanger Institute, Wellcome Genome Campus, Cambridge, UK. [13]Interdepartmental Program in Computational Biology and Bioinformatics, Yale University, New Haven, CT, USA. [14]Konrad Zuse School of Excellence in Learning and Intelligent Systems (ELIZA), Darmstadt, Germany. [15]Helmholtz AI, Helmholtz Zentrum München, Munich, Germany. [16]Wellcome MRC Cambridge Stem Cell Institute, University of Cambridge, Cambridge, UK. [17]Institute of Experimental Pneumology, LMU University Hospital, Ludwig-Maximilians University, Munich, Germany. ✉e-mail: fabian.theis@helmholtz-muenchen.de

## Methods

### Implementation of pertpy

Pertpy is implemented in Python and builds upon several scientific open-source libraries, including NumPy[70], Scipy[71], JAX[15], scikit-learn[72], Pandas[72,73], AnnData[22], scanpy[34], muon[23], NumPyro[74], OTT-JAX[75], blitzG-SEA[69], PyTorch[76] and scvi-tools[13] for omics data handling and matplotlib[77] and seaborn[78] for data visualization.

**Summary table of implemented methods.** Pertpy provides implementations of many novel, but also established, methods that can be easily accessed and combined to easily build custom analysis pipelines (Table 1).

**gRNA assignment.** Assigning relevant guides to each cell is essential in genetic perturbation assays, ensuring that the observed cellular responses are accurately linked to the intended genetic modifications. This step is critical for validating experimental design and interpreting results reliably. Pertpy provides two approaches to assigning cells to guides.

First, a simple thresholding model where the most expressed gRNA is assigned to a cell if it additionally exceeds an optional user-specified count threshold.

Second, a previously published Poisson–Gaussian model[11]. For each guide, cells with non-zero expression are $\log_2$ transformed and modeled as a mixture of two populations, with cells automatically classified as negative if they show zero expression. A cell is labeled as positive for a guide if it belongs to the higher-expressing population, with a maximum of five guide assignments per cell to prevent over-assignment; cells exceeding this threshold are marked as 'multiple', whereas those failing to meet the mixture model threshold for any guide are designated as 'negative'.

**Differential gene expression.** Differential gene expression analysis compares the mean gene expression levels between different conditions or groups to identify genes with statistically significant changes, utilizing statistical models to account for between-sample variability and control for false discovery rates. Pertpy provides a unified application programming interface (API) to support a variety of such models. The first group of models comprises the *t*-test and Wilcoxon test as simple statistical tests for comparing expression values between two groups without covariates. The second group includes models of the linear model family that allow modeling complex designs and contrasts. Currently included are PyDESeq2[35], edgeR[36] as well as a wrapper around statsmodels (https://www.statsmodels.org), which provides access to a wide range of regression models, including ordinary least squares regression, robust linear models and generalized linear models. Linear model designs can be specified via Wilkinson formulas as known from R (through 'Formulaic', https://github.com/matthewwardrop/formulaic). Pseudobulk workflows that account for pseudoreplication bias[79] are enabled by integration with scanpy's get.aggregate() function. Results tables ranked by adjusted *P* value are provided as a Pandas data frame and can be visualized using volcano plots.

**Analysis of pooled CRISPR screens with mixscape.** CRISPR–Cas9 can sometimes lead to cells escaping gene perturbation, such as knockout, by receiving an ineffective in-frame mutation, underscoring the necessity for computational quality control to predict and enhance their specificity and performance. Mixscape classifies targeted cells—that is, those identified as perturbed by presence of a gRNA—into successfully perturbed (KO) and targeted but not successfully perturbed (NP) based on their response. Other perturbations, such as activations or inhibitions, are here collectively referred to as 'KO' for consistency with the original publication.

In particular, the Mixscape pipeline includes the following steps:

(1) Calculate the perturbation-specific signature of every cell, which is the difference of the targeted and the closest *k* (defaults to 20) nearest control neighbors.

(2) Identify and remove cells that have 'escaped' CRISPR perturbation by estimating the distributions of KO cells. Afterwards, the posterior probability that a cell belongs to the KO cells is calculated, and the cells are binary assigned based on a fixed probability threshold (defaults to 0.5).

(3) Visualize similarities and differences across different perturbations using linear discriminant analysis.

When calculating the perturbation-specific signatures, Mixscape makes strong assumptions, such as cells with a perturbation not exhibiting compositional differences with respect to variation seen within the control cells. Additional limitations include the assumption that perturbation effects are additive and separable from underlying cell state, the equal weighting of all genes regardless of their relevance to the perturbation target and the failure to account for temporal dynamics in cellular responses where early and late responding genes create composite signatures.

Generally, the Mixscape pipeline assumes KO data. Applying Mixscape to CRISPR interference (CRISPRi) and CRISPRa data is more nuanced but still valid under certain conditions. Unlike KO, these modalities do not introduce permanent genomic alterations, but variability in perturbation efficiency can create functionally not effectively perturbed cells. Factors such as incomplete transcriptional repression/activation, gRNA efficiency, chromatin state, CRISPR expression or variable effector recruitment (for example, KRAB for CRISPRi and VP64 for CRISPRa) can lead to heterogeneous perturbation effects. If these effects result in a clear separation between perturbed and unperturbed-like transcriptomic states, Mixscape can still be meaningfully applied. However, careful validation is needed to ensure that the identified unperturbed population reflects true biological variability rather than technical artifacts.

We implemented Mixscape following the implementation of the original authors[19]. We further optimized the implementation by using PyNNDescent (https://github.com/lmcinnes/pynndescent) for nearest neighbor search for the calculation of the perturbation signature.

The implementation was verified by comparing the classification results between the original Seurat Mixscape implementation and the pertpy implementation through a confusion matrix, showing high agreement, with 4,674 KO, 13,098 NP and 2,386 non-targeted cells correctly classified by both implementations, with only minor disagreements (438 cells classified as NP by pertpy but KO by original and 133 cells classified as KO by pertpy but NP by original). Additionally, the perturbation signature scores between implementations show a strong correlation of 0.97 (*P* < 0.0001), confirming that pertpy's implementation closely reproduces the original method's quantitative measurements.

**Compositional analysis of labeled groups with scCODA and tascCODA.** Tracking cell type shifts is crucial for understanding the underlying mechanisms of disease progression, tissue regeneration and developmental biology, offering insights into cellular responses and adaptations. Despite their critical role in biological processes such as disease, development, aging and immunity, detecting shifts in cell type compositions through scRNA-seq is challenging. Statistical analyses must navigate various technical and methodological constraints, including limited experimental replicates and compositional sum-to-one constraints[37]. scCODA and its extension tascCODA both employ Bayesian methods to elucidate cell type compositional changes, with tascCODA being able to also take cell type hierarchies into account.

The implementations of scCODA 2.0 and tascCODA 2.0 are mathematically equivalent to the original implementations[37,38] but allow for accelerated inference by replacing the Hamiltonian Monte Carlo algorithm with the no-U-turn sampler from NumPyro[74]. The joint implementation also allows users to conveniently apply both methods from within the same framework.

Pertpy further uses MuData[23] objects to simultaneously handle cell-by-gene and sample-by-cell-type representations of the same data, simplifying the data aggregation and model specification steps for scCODA 2.0 and tascCODA 2.0 while ensuring compatibility with other methods featured in the scverse[14] ecosystem. A wide range of visualization options through scanpy[34], ETE 3 (ref. [80]) and ArviZ[81] for representation of differentially abundant cell types, their hierarchical structure and inference diagnostics, respectively, are also provided within pertpy.

The implementation was verified by comparing parameter estimates and log₂ fold changes with the original implementation across multiple test scenarios, including different reference cell types and treatment conditions, with results showing nearly identical values between implementations (within approximately 0.01 for parameters and approximately 0.005 for $\log_2$ fold changes).

**Compositional analysis of unlabeled groups with Milo.** Most methods for comparing single-cell datasets often rely on identifying discrete clusters to test for differences in cell abundance across experimental conditions. However, this approach may lack the necessary resolution and fail to represent continuous biological processes accurately. To address these limitations, Milo was designed to conduct differential abundance tests by assigning cells to overlapping neighborhoods within a $k$-nearest neighbor graph.

The implementation of Milo is based on Milopy (https://github.com/emdann/milopy). It uses the same MuData-based data structure that the scCODA 2.0 and tascCODA 2.0 implementations also use. Here, neighborhood counts are stored in a slot in MuData for downstream usage.

The implementation was verified by comparing the results from the pertpy implementation and the original miloR package, showing a strong correlation ($r = 0.987$) between log fold change values calculated at the cell level. Additionally, precision and recall analysis across different significance thresholds demonstrated high concordance between the two implementations, with both metrics approaching 1.0 as the threshold increases. This confirms that pertpy's Milo implementation accurately reproduces the statistical findings of the original method.

**MCPs with DIALOGUE.** MCPs, or gene programs, refer to the complex regulatory networks and signal transduction pathways that govern the behavior, differentiation and communication of cells. DIALOGUE[40] is a matrix factorization method for identifying these specific gene expression patterns. The implementation of DIALOGUE in pertpy resembles the original implementation[40]. The main differences are as follows:

- The R implementation of MultiCCA has been replaced with a Python implementation of the original mathematical formulation[82], which can be found at https://github.com/theis-lab/sparsecca. In addition, the Python implementation also has the option to solve for the canonical covariate weights $w$ using linear programming, allowing for concurrent instead of iterative optimization over the pairwise factor matrices. This results in weights that are consistent regardless of the order in which cell types are passed, which was not previously true.
- An additional gene identification method, referred to as extrema MCP genes, which selects cells at the extreme values of the MCP (cells with the top 10% and bottom 10% MCP scores in each cell type) and then runs the rank_genes_groups function from scanpy with default parameters to perform a $t$-test between the two groups of cells to identify differentially expressed genes to provide adjusted $P$ values based on the number of tested genes.

Although the extrema MCP genes approach utilizes gene expression data twice—once for defining MCPs and again for differential testing—it avoids statistical circularity common in post-clustering analyses[83]. Unlike traditional clustering approaches where cells are forcibly separated based on expression patterns and then the same data are used to identify what drives that separation (creating artificially small $P$ values), the MCP scores represent continuous axes of biological variation extracted through independent matrix factorization methods, whereas the extrema selection merely applies thresholds to these pre-computed scores. The subsequent differential expression testing therefore examines distinct biological phenomena rather than confirming the same signal, maintaining statistical validity and interpretability of the identified gene signatures.

Owing to these differences, the reported MCPs and MCP genes will not exactly match those identified in the DIALOGUE R package. Notably, users should be aware that the Seurat and scanpy implementations calculate principal component analysis (PCA) differently, resulting in downstream differences in MCP scores. When the same PCA representation is used, the MCP values between the R and Python implementation have an average Pearson's correlation of 0.96 when tested on the sample dataset provided in the R tutorial.

**Enrichment with blitzGSEA.** Gene set enrichment analysis (GSEA) determines whether predefined sets of genes, often associated with specific biological functions or pathways, show statistically significant, concordant differences in expression across two biological states or phenotypes. It is used to identify biological processes that are overrepresented in a ranked list of genes, typically arising from high-throughput experiments. This approach shifts the analysis focus from individual genes to the collective behavior of genes within predefined, functionally related groups, facilitating a deeper understanding of the biological mechanisms underlying observed changes. Pertpy provides access to a variety of metadata databases that provide gene sets whose enrichment can be tested for.

We generally followed the enrichment pipeline described in Drug2Cell[66] to test for the enrichment of gene sets. This pipeline entails:

(1) Fetching gene sets from databases
(2) Scoring gene sets by computing the mean expression of each gene group per cell
(3) Performing a differential expression test to get ranked gene groups that are upregulated in particular clusters
(4) Determining enriched genes using a hypergeometric test on the gene set scores or using blitzGSEA[69]

The implementation was verified by comparing the results from pertpy's enrichment module and the original Drug2Cell package, demonstrating exact equivalence in both overrepresentation and enrichment analyses. Tests confirmed that the pertpy implementation produces identical results for hypergeometric overrepresentation testing in cell-type-specific pathways and GSEA, with all results being equal.

**Distances, metrics and permutation tests.** Distance metrics serve as an important baseline in two primary tasks in single-cell perturbation analysis: (1) identifying relative heterogeneity and response and (2) evaluating and training single-cell perturbation models. To this end, various commonly used distance metrics have been implemented to be easily applied to single-cell AnnData objects with accompanying perturbation or disease labels. In the following, we present the 16 distances, in order of performance according to Ji et al.[59], that are implemented in pertpy. We use $x^k$ to denote the gene expression in cell $k$ and $x_i$ and $y_i$ for the expression of gene $i$ in the perturbed and control conditions, respectively.

- **MSE**
  Determines the mean squared distance between the mean vectors of two groups.

$$\text{MSE} = \frac{1}{n} \sum (x_i - y_i)^2$$

- **Maximum mean discrepancy (MMD)**
  Evaluates the discrepancy between the empirical distributions of two groups using kernel-based methods. Let $n$ denote the number of samples and $k(\cdot, \cdot)$ the linear kernel function.

$$\text{MMD}^2 = \frac{1}{N(N-1)} \sum_{i=1}^{N} \sum_{j \neq i}^{N} k(x^i, x^j) - \frac{2}{NM} \sum_{i=1}^{N} \sum_{j=1}^{M} k(x^i, y^j)$$

$$+ \frac{1}{M(M-1)} \sum_{i=1}^{M} \sum_{j \neq i}^{M} k(y^i, y^j)$$

- **Euclidean distance**
  Calculates the Euclidean distance between the means of the two groups.

$$\text{Euclidean distance} = \sqrt{\Sigma(x_i - y_i)^2}$$

- **Energy distance**[11,43]
  Computes a statistical energy distance between two groups based on mean pairwise distances within and between groups. We define

$$\delta_{XY} = \frac{1}{NM} \sum_{i=1}^{M} \sum_{j=1}^{N} \|x^i - y^j\|,$$

$$\delta_X = \frac{1}{N(1-N)} \sum_{i=1}^{N} \sum_{j=1}^{N} \|x^i - x^j\|$$

and $\delta_Y$ accordingly, where $\delta$ denotes the mean pairwise distance between samples. The energy distance is then calculated as

$$E(X, Y) = 2\delta_{XY} - \delta_X - \delta_Y$$

- **Kolmogorov–Smirnov test distance**
  Applies the Kolmogorov–Smirnov statistic to measure the maximum distance between the empirical cumulative distributions of two groups. We define the empirical distribution function for gene $i$ as

$$f_i(z) = |\{y_i^k : y_i^k \leq z, k \in \{1, \dots, N\}|$$

over all cells of the control condition and, analogously, $\hat{f}_i(z)$ for perturbed cells. For each gene, the maximum distance between both distribution functions $\max_{z \geq 0} |f_i(z) - \hat{f}_i(z)|$ is computed, and the results are averaged over all genes to yield a single distance value.

- **Mean absolute error (MAE)**
  Measures the mean absolute difference between the mean vectors of two groups.

$$\text{MAE} = \frac{1}{n} \sum |x_i - y_i|$$

- **Two-sided $t$-test statistic**
  Uses the $t$-test statistic to compare the means of two groups under the assumption of unequal variances. Let $s_{x_i}^2$ and $s_{y_i}^2$ denote the variances of gene $i$ for perturbed and control, $n_x$ and $n_y$ the sample sizes for perturbed and control and $\epsilon$ a small factor to avoid dividing by zero.

$$t = \frac{1}{n} \sum \frac{x_i - y_i}{\sqrt{\frac{s_{x_i}^2}{n_x + \epsilon} + \frac{s_{y_i}^2}{n_y + \epsilon}}}$$

- **Cosine distance**
  Computes the cosine of the angle between the mean vectors of the two groups.

$$\text{Cosine distance} = 1 - \frac{x \cdot y}{|x| \cdot |y|}$$

where $\cdot$ denotes the dot product.

- **Pearson's distance**
  Uses Pearson's correlation to assess the linear correlation between the mean vectors of two groups, returning 1 minus the correlation coefficient. Let $\underline{x}$ and $\underline{y}$ denote the mean expression over all genes.

$$r = 1 - \frac{\sum(x_i - \underline{x})(y_i - \underline{y})}{\sqrt{\sum(x_i - \underline{x})^2 \sum(y_i - \underline{y})^2}}$$

- **Coefficient of determination distance**
  Calculates the coefficient of determination ($R^2$) between the mean vectors of two groups. Note that, unlike most other distances listed here, $R^2$ is not symmetric/has not been symmetrized.

$$R^2 = \frac{\sum(x_i - y_i)^2}{\sum(x_i - \underline{x})^2}$$

where $\underline{x}$ is the mean expression over all genes in the perturbed condition.

- **Classifier control probability**
  To compute the classifier class projection distance between perturbations $P$ and control condition $C$, we train a linear regression classifier to distinguish between $C$ and $P$, with 20% of $P$ held out for testing. To calculate the distance for perturbation class $P_i$, we obtain the average post-softmax classification probabilities of all cells in $P_i$ and return the probability of class $C$.

- **Kendall's tau distance**
  Applies Kendall's tau, a measure of ordinal association, between the mean vectors of two groups. We define $C$ as the number of concordant pairs, $D$ as the number of discordant pairs, $X$ as the number of ties in $x$'s ranking and $Y$ as the number of ties in $y$'s ranking.

$$\tau'_{xy} = \left(1 - \frac{(C-D)}{\sqrt{(C+D+X)(C+D+Y)}}\right) \frac{n(n-1)}{4}$$

- **Spearman's rank distance**
  Similar to Pearson's distance but uses Spearman's rank correlation to measure nonlinear relationships.

$$\rho = \frac{6 \Sigma d_i^2}{n(n^2 - 1)}$$

where $d_i$ represents the difference in rank of gene $i$ across both samples.

- **Wasserstein distance**
  Also known as Earth Mover's Distance, computes the cost of optimally transporting mass from one distribution to another. Let $W(p, q)$ be the first-order Wasserstein distance between probability distributions $p$ and $q$, $\Gamma(p, q)$ the set of all joint distributions with marginals $p$ and $q$ and $c(x, y)$ the cost of transporting a unit of mass from $x$ to $y$, and $X$ and $Y$ are the support sets of $p$ and $q$, respectively.

$$W(p, q) = \inf_{\gamma \in \Gamma(p,q)} \int_{X \times Y} c(x, y) d\gamma(x, y)$$

- **Symmetric Kullback–Leibler divergence**
  Measures how one probability distribution diverges from a second. In the case of discrete inputs, the Kullback–Leibler divergence is calculated as follows:

$$D_{\text{KL}}(P\|Q) = \sum_{x \in \Omega} P(x) \log\left(\frac{P(x)}{Q(x)}\right)$$

where $P$ and $Q$ are discrete probability distributions. For non-discrete inputs, the Kullback–Leibler divergence is computed as

$$KL = \sum \ln \frac{s_{y_i}}{s_{x_i}} + \frac{s_{x_i}^2 + (x_i - y_i)^2}{2 * s_{y_i}^2} - \frac{1}{2}$$

where $s$ denotes the standard deviation.

- **Classifier class projection**
  The classifier class projection distance between perturbation $P_i$ and control condition $C_i$ is calculated by training a linear regression classifier on all $x \notin P_i$ and all $C$, subsequently retrieving the average post-softmax classification probabilities of all cells $\underline{x}_i$ and returning the probability of class $C_i$.

  The following distance was also implemented in pertpy but was not part of the aforementioned benchmark:

- **Negative binomial log likelihood**
  Fits a negative binomial distribution to one group and uses it to compute the log likelihood of the other group's data. For each gene $i$ that is not overdispersed in $x$, we fit a negative binomial distribution with parameters $\mu_i$ and $\theta_i$. The distance between two categories $x$ and $y$ is then computed as the average negative log likelihood of $y$ given the parameters of the distribution fit on $x$ for each gene $i$—that is,

$$1/n \sum_{i=1}^{N} \theta_{x_i}(\log(\theta_{x_i}) - \log(\theta_{x_i} + \mu_{x_i})) + y_i(\log(\mu_{x_i}) - \log(\theta_{x_i} + \mu_{x_i}))$$
$$+\ln(\Gamma(y_i + \theta_{x_i})) - \ln(\Gamma(\theta_{x_i})) - \ln(\Gamma(y_i + 1))$$

The 'distances' module allows users to quickly fetch the pairwise distances between any set of categorically labeled cells. The 'distance_tests' module allows users to compute a $P$ value through Monte Carlo permutation testing, thereby providing a confidence value for any given distance. This can be particularly comforting in cases in which distances have been used as proxies for real biological response in gene expression space.

Note that, although we refer to all of the above as 'distances', they do not all meet the mathematical definition of a distance; deviations from the standard distance axioms are detailed in Ji et al.[59]. Although these distances can be used with any single-cell measurement, it should be noted that the ranking above was performed in the context of single-cell transcriptomics.

We also implemented two metrics for evaluating expression prediction models. To evaluate if perturbation prediction leads to meaningful biological conclusions, we implemented a differential expression correlation metric. This metric uses Spearman's correlation to compare differential gene ranking from the scanpy rank_genes_groups function performed on control versus real perturbed data and on control versus predicted perturbed data. To evaluate if the distribution of gene expression means versus variances corresponds to real data, we used a similar method as proposed previously[84]. The distribution of expression mean–variance two-dimensional relationship was estimated with kernel density for both real and predicted perturbed data. The distance between the two densities was estimated based on the difference of values sampled across the whole data range.

**Perturbation ranking with Augur.** Augur aims to rank or prioritize cell types according to their response to experimental perturbations. The fundamental idea is that, in the space of molecular measurements, cells reacting heavily to induced perturbations are more easily separated into perturbed and unperturbed than cell types with little or no response. This separability is quantified by measuring how well experimental labels (for example, treatment and control) can be predicted within each cell type. Augur trains a machine learning model predicting experimental labels for each cell type in multiple cross-validation runs and then prioritizes cell type response according to metric scores measuring the accuracy of the model. For categorical data, Augur uses the AUC, and, for numerical data, it uses the concordance correlation coefficient.

Our implementation of Augur follows the original implementation[67,68]. We further optimized it by parallelizing the training of the predictive models. Moreover, the pertpy implementation allows for gene selection using either the originally used variance based implementation or scanpy's highly variable genes.

The implementation was verified by comparing the results from pertpy's Augur implementation and the original R-based Augur package, showing excellent agreement in both default and velocity mode. The AUC scores from both implementations were highly consistent across all tested cell types, with all data points falling within 4% of the expected $y = x$ line. This close correspondence was observed in both analysis modes, confirming that pertpy's implementation faithfully reproduces the computational methodology of the original R package.

**Causal identification of single-cell experimental perturbation effects with CINEMA-OT.** Cellular responses to environmental signals are crucial for understanding biological processes. Effectively extracting biological insights from such data, especially through single-cell perturbation analysis, remains challenging due to a lack of methods that can directly account for underlying confounding variations. CINEMA-OT distinguishes between confounding variations and the effects of perturbations, achieving an optimal transport match that mirrors counterfactual cell pairings. These pairings allow for the analysis of causal perturbation responses, enabling novel approaches, including individual treatment effect analysis, clustering of responses, attribution analysis and the examination of synergistic effects.

The implementation of CINEMA-OT is based on the original implementation[41]. We used OTT-JAX[79] to make the implementation portable across hardware. It can, therefore, also be run on GPUs. Notably, the JAX-based implementation may initially run slower than the NumPy-based version due to the overhead of just-in-time compilation.

The implementation was verified by comparing the results from pertpy's CINEMA-OT implementation and the original CINEMA-OT package. Tests showed strong agreement between both implementations, with a relative Frobenius norm difference of less than 0.1 (0.0973) for the optimal transport transformed confounders. Additionally, single-cell treatment effects showed exceptionally high correlation between implementations, with mean Pearson's correlation of 0.989 and mean Spearman's correlation of 0.983 across all genes. Both implementations consistently revealed the same biological insight regarding distinct treatment effects in monocytes, confirming that pertpy's implementation faithfully reproduces the computational methods of the original tool.

**Perturbation spaces.** Pertpy discriminates between two fundamental domains to embed and analyze data: the 'cell space' and the 'perturbation space'. In this paradigm, the cell space represents configurations where discrete data points represent individual cells. Conversely, the perturbation space departs from the individualistic perspective of cells and, instead, categorizes cells based on similar response to perturbation or expressed phenotype where discrete data points represent individual perturbations. This specialized space enables comprehending the collective impact of perturbations on cells. We differentiate between perturbation spaces (where we create one data point for all cells of one perturbation) and cluster spaces (where we cluster all cells and then test how well the clustering overlaps with the perturbations).

**Pseudobulk space.** This space takes the pseudobulk of a covariate such as the condition to represent the respective perturbations using the Python implementation of DecoupleR[44] (https://github.com/saezlab/decoupler-py), which can subsequently be embedded.

**Centroid space.** The centroid space calculates the centroids as the mean of the points of a condition for a pre-calculated embedding. Next,

it finds the closest actual point to that centroid, which determines the perturbation space point for that specific condition.

**MLP classifier space.** The MLP classifier space trains a feed-forward neural network to predict which perturbation has been applied to a given cell. By default, a neural network with one hidden layer of 512 neurons and batch normalization is created and trained using a batch size of 256. However, all these hyperparameters can be customized by the user to suit the specific requirements of the dataset. We account for class imbalances by oversampling perturbations with fewer instances. The MLP is trained using cross-entropy loss until detection of overfitting (early stopping) or until it reaches the maximum number of epochs to train, set to 40 by default. To obtain perturbation-informed embeddings of the cells, the cell representations in the last hidden layer are extracted. Another perturbation space, such as pseudobulk, can be applied downstream to obtain a per-perturbation embedding if required. For creation and training of the MLP, we leverage the PyTorch library.

**Logistic regression classifier space.** The logistic regression classifier space generates perturbation embeddings, as opposed to per-cell embeddings computed by the MLP classifier space. A logistic regression classifier is trained for each perturbation individually to determine if the respective perturbation was applied to a cell or not. Depending on user preference, the classifier can be trained on the high-dimensional feature space or on a pre-computed embedding, such as one obtained through PCA. For each perturbation, we extract the coefficients of the logistic regression classifier, trained until convergence or reaching the maximum number of iterations (1,000 by default), to derive a per-perturbation embedding. We use scikit-learn's implementation for the logistic regression classifier.

*DBSCAN space.* DBSCAN[85] (density-based spatial clustering of applications with noise) is a clustering algorithm that identifies clusters in a dataset based on the density of data points, grouping together points that are closely packed while marking points in low-density regions as outliers. Pertpy's implementation of a DBSCAN space is based on scikit-learn's DBSCAN implementation.

**$k$-means space.** $k$-means is a clustering algorithm that partitions a dataset into $k$ distinct, non-overlapping clusters by minimizing the distance between data points and the centroid of their assigned cluster. It iteratively adjusts the positions of centroids to reduce the total variance within clusters, making it suitable for identifying spherical-shaped clusters in feature space. Pertpy's implementation of a $k$-means space uses $k$-means clustering as implemented in scikit-learn.

**Label transfer.** Label transfer in single-cell analysis involves using annotations of a dataset to predict the states of unannotated data points, leveraging similarities in gene expression patterns or nearest neighbors. Pertpy's label transfer function uses PyNNDescent to find the closest neighbors for all data points and then uses majority voting to label unlabeled data points.

The label transfer function further quantifies uncertainty, where each neighbor's contribution is weighted by its connectivity strength (derived from the distance in gene expression space). These weighted contributions are first converted into a one-hot encoded matrix where each column represents a label category. The uncertainty score for each transferred label is then calculated as the Shannon entropy of the weighted label distribution in the cell's neighborhood—if all neighbors have the same label, the entropy (and, thus, uncertainty) is 0, whereas diverse labels among neighbors result in higher entropy values. This uncertainty score provides a quantitative measure of prediction confidence, where higher values indicate more heterogeneous neighborhoods and, thus, less reliable label transfers.

Any obtained labels through label transfer must be diligently verified. Label transfer can propagate biases from the reference annotations, leading to incorrect annotations if the reference is not representative of the target data. Differences in batch effects, technical noise or biological variability can distort nearest neighbor relationships, reducing the reliability of transferred labels. Additionally, majority voting can fail in cases where distinct perturbations and cell states are underrepresented, leading to misclassification of rare populations.

**Metadata support.** Pertpy provides access to several databases that contain additional metadata for cell lines, mechanisms of actions and drugs. On request, the database content gets cached locally, and the respective information gets stored in the appropriate slots of the passed AnnData object.

**Cell line.** Pertpy provides access to DepMap (https://depmap.org/portal/, version 23Q4) and GDSC[29]. The following information can be obtained:

- **Cell line identification**: Comprehensive details such as cell line names, aliases, DepMap IDs and CCLE[86] names
- **Genetic information**: Data on genetic aberrations prevalent in cancer cell lines, including mutations, copy number alterations, fusion genes and comprehensive gene expression profiles
- **Dependency scores**: Quantitative assessments of gene essentiality that showcases the impact of specific genes on the viability of cancer cell lines
- **Drug sensitivity**: Detailed measurements of how cancer cell lines respond to various drugs, with metrics such as half-maximal inhibitory concentration ($IC_{50}$) values providing insights into the effectiveness and potential toxicity of therapeutic compounds
- **Lineage and type**: Information categorizing cell lines based on their tissue of origin and the type of cancer they represent
- **Molecular subtypes**: Classifications based on detailed genetic, epigenetic and proteomic analyses, which help in understanding the heterogeneity within and across cancer types
- **Phenotypic data**: Observations on cell growth rates and morphological characteristics, which can correlate with genetic traits and drug responses
- **Genomic profiling**: Includes high-resolution data from whole-exome and whole-genome sequencing efforts, offering a comprehensive view of the genetic landscape of cell lines
- **Proteomics profiling**: Protein intensity values acquired using data-independent acquisition mass spectrometry (DIA-MS) from DepMap Sanger.

**Mechanism of action.** Pertpy provides access to CMAP[30], also commonly referred to as CMap and LINCS Unified Environment (CLUE), which hosts the infrastructure. CMAP is a resource designed to help researchers discover functional connections among diseases, genetic perturbation and drug action. The following information can be obtained:

- **Compound names**: The name of the compound of genetic perturbagen
- **Mechanism of action**: The specific biochemical interactions through which compounds exert their effects on cellular functions. This includes detailed descriptions of whether a compound acts as an inhibitor, activator or modulator of particular molecular targets.
- **Target**: The sets of genes or proteins that directly interacted with or were affected by the perturbagen

**Drug.** Pertpy provides access to PubChem[31] using PubChemPy (https://github.com/mcs07/PubChemPy). PubChem is a comprehensive resource for chemical information, primarily known for its vast database of chemical molecules. The following information can be obtained:

- **Chemical identifiers:** Each chemical in PubChem is assigned unique identifiers, including CAS numbers, InChI strings and SMILES notation.
  Pertpy further provides access to the ChEMBL[32] database. ChEMBL is a comprehensive database maintained by the European Bioinformatics Institute, part of the European Molecular Biology Laboratory. It provides a vast collection of data on bioactive molecules with drug-like properties. The following information can be fetched:
- **Compounds:** The names of the compounds
- **Targets:** The target gene sets of the compounds

**Benchmarking runtime.** To evaluate computational efficiency, we measured execution time and resource consumption for all tools implemented in pertpy. Following their respective tutorials, we developed scripts with standard workflows on exemplary data in their original implementation. We further wrapped all of these scripts in a reproducible Snakemake[87] pipeline using Conda environments that we defined per tool implementation to create isolated and reproducible runtime environments.

These scripts were executed on a system with an AMD EPYC 9754 128-core processor and 500 GB RAM in a Linux environment. This setup ensured accurate and reproducible timing measurements. Each script was run three times to guarantee consistency. We upsampled or downsampled example datasets with a set seed to evaluate each tool at 5,000, 10,000, 50,000, 100,000, 500,000 or 1,000,000 cells. Timing and memory use was recorded with Snakemake's benchmark feature. The results are shown in a box plot (Extended Data Fig. 1), which compares the execution time in seconds and memory usage in megabytes across each tool and implementation.

## Use cases

For the following analyses, we used the latest pertpy version (0.10.0). We deposited a full Conda environment to reproduce our results in the associated reproducibility repository, together with all result tables of our analysis.

**Analysis of the CRISPR screen dataset.** We obtained the original dataset from the original publication[16], together with the labels of the gene programs. The dataset contained 111,255 cells and 19,018 genes. We followed the standard scanpy preprocessing pipeline to log normalize the data, calculate 4,000 highly variable genes, obtain PCA components and embed the data into a uniform manifold approximation and projection (UMAP) space for visualization purposes. Moreover, we scored cell cycle genes using the list of Tirosh et al.[88].

Afterwards, we compared three distinct processing strategies: (1) perturbation signature computation and cell filtering based on the 20 nearest neighbor control cells, (2) perturbation signature computation and cell filtering based on all control cells within the same gene group and (3) no perturbation-signature-based cell filtering. For strategies (1) and (2), we used pertpy's implementation of Mixscape to calculate the perturbation signature (with ref_selection_mode = 'nn' for strategy (1) and ref_selection_mode = 'split_by' for strategy (2) in pt.tl.Mixscape. perturbation_signature), which was subsequently embedded into UMAP space. Next, we applied Mixscape to the perturbation signature to calculate the perturbation scores that are binarized to label cells as successfully and unsuccessfully perturbed.

We applied pertpy's MLP-based classifier to the gene expression data from each processing strategy (with cells filtered out for strategies (1) and (2)) and embedded the pseudobulk of the penultimate layer feature values with UMAP. To quantify the similarity of the perturbation spaces produced by each processing strategy, we used scikit-learn to calculate the silhouette score for each perturbation from the UMAP of the perturbation space. We then averaged the silhouette scores for each gene program. The silhouette score varies between −1 and +1, where a higher score indicates that the perturbation embedding is well aligned with its corresponding gene program cluster and poorly aligned with other gene program clusters.

We further used pertpy's distance module to compute the MSE distance between the two subclusters formed from perturbations annotated as pro-growth. To assess the importance of individual genes (input features) for predicting perturbations, we calculated integrated gradients[46] using captum (captum.attr.IntegratedGradients). We computed the attribution for each cell using its respective perturbation label as the target and then averaged the feature importances across all cells annotated with the same gene program.

To identify gene programs affected by perturbations in an unannotated cluster in the UMAP, we performed GSEA on either upregulated or downregulated genes (adjusted *P* value cutoff of 0.01) in the cluster of interest, identifying the top three upregulated and downregulated Reactome[89] pathways for the cluster.

**Analysis of the chemical perturbation dataset.** We obtained the dataset from the original publication of the study, which already contained annotations of cell lines, cell line quality, channel, disease, dose units, dose values and many more fields that are documented in our analysis notebook. We filtered out cells perturbed by CRISPR, leaving 154,710 cells and 32,738 genes of 172 cell lines, treated with 13 different drugs. We applied standard preprocessing by filtering genes that were present in fewer than 30 cells and log normalizing the counts. In total, 4,000 highly variable genes were computed using the highly_variable_genes function of scanpy and used as the basis for downstream analyses, except when examining viability-dependent and viability-independent drug responses.

Next, we fetched all available cell line metadata from DepMap and GDSC, using pertpy to annotate the cell lines by their DepMap ID with cell lineages, compound targets and mechanism of action using CMAP[30]. We further added drug sensitivities of cell lines to anticancer therapeutics from GDSC[29] and PRISM (DepMap).

Pseudobulks were generated using pertpy's PseudobulkSpace function by perturbation. We used the expression of the cell lines labeled as 'control' as baselines. Bulk RNA expression data were fetched from the CCLE using the data from the Broad Institute via pertpy. We used pertpy to calculate row-wise correlations of the expression profiles of the cell lines to obtain Pearson's correlation values and *P* values.

Finally, we used pertpy to disentangle drug responses into components that are independent of and dependent on the sensitivity of a certain cell line to a drug. We followed the approach presented in the paper introducing the original dataset[17] but replaced functionalities with pertpy's own implementation whenever possible. Although previous work focused on the drug trametinib, we here investigated treatment responses to dabrafenib. We used pertpy's annotate_from_gdsc function to query the AUC values for each cell line−drug combination using the GDSC1, GDSC2 and PRISM databases. We rank normalize the AUC values within each database and then compute the mean of all available values for)each cell line. The dabrafenib sensitivity is then defined as 1 minus the mean AUC. Next, we computed the expression log fold change between treated cells and control based on raw counts for each cell line individually, using pertpy's implementation of edgeR. Then, for each gene, the following linear regression model was fit:

$$\log-FC_{Gene} = Intercept + Slope \times Dabrafenib\ sensitivity\ of\ cell\ lines$$

The fit model enables the decomposition of the observed change in gene expression in the treatment group into two components: a viability-independent response (intercept) and a viability-dependent response (slope). Genes with a Benjamini−Hochberg-corrected *P* value less than 0.01 for either the slope or intercept were considered significant and subsequently used for GSEA using the blitzGSEA[69] API.

**Analysis of the TNBC treatment dataset.** We obtained the dataset from the original publication[18], which comprises scRNA-seq and assay for transposase-accessible chromatin with sequencing (ATAC–seq) data from 22 patients with advanced TNBC, treated with paclitaxel alone or in combination with the anti-PD-L1 therapy atezolizumab. We focused on the transcriptomic data that encompass 489,490 high-quality immune cells with 27,085 measured genes across 99 high-resolution cell types. We restricted the dataset to tumor biopsy samples (excluding peripheral blood) and included only patients who exhibited either a partial response to treatment or stable disease (excluding one patient with progressive disease), resulting in a final cohort of 15 patients. We filtered genes with fewer than 10 cells, log normalized the data and selected highly variable genes using scanpy defaults. We calculated a PCA representation using scanpy with default settings that uses the 'arpack' solver. For the following analyses, we filtered the dataset to only keep cell types that were retained in all response groups.

To determine compositional changes, we applied pertpy's implementation of scCODA per treatment. scCODA's automatic reference cell type detection determined intermediate monocytes as the reference cell type, which we used for both treatments for consistency. Compositional changes with a false discovery rate of 0.1 (10%) were marked as credible effects.

We calculated the MSE distance between the respective groups in a pairwise fashion using pertpy's 'distance' module on the PCA representation. We repeated this process three times for both treatments jointly, only chemotherapy treatment and only anti-PD-L1 and chemotherapy combination treatment.

DIALOGUE decomposition analysis was carried out exclusively on pre-treatment tumor samples. The sample labeled 'Pre_P010_t' was excluded because it demonstrated low diversity in cell types. The analysis was confined to cell types that had a minimum of three cells per sample in the remaining patient samples. The number of MCPs was to set 10, with normalization enabled and the 'LP' solver. We pooled patients receiving both treatments for this analysis, as DIALOGUE requires that all cell types analyzed be present in all patients.

When testing for associations between MCPs and treatment response, we applied a hierarchical testing approach by first examining cell types within each MCP individually. A predictive MCP for treatment response was determined using a $t$-test for independent samples for each cell type within each MCP. To adjust for the number of cell types tested, the Benjamini–Hochberg correction method was applied. Although we corrected for multiple testing across cell types, we acknowledge that additional correction across all MCPs would be more conservative. We chose this approach to balance statistical stringency with the exploratory nature of our analysis, as overly conservative correction might obscure biologically meaningful patterns in this high-dimensional dataset with limited sample size. The biological relevance of our findings is further supported by the consistent directionality of MCP2 effects across multiple functionally distinct immune cell populations, an outcome highly unlikely to occur by chance alone. Instead, we opted for more stringent thresholds in subsequent gene-level analyses, where we identified significantly associated genes with extremely low adjusted $P$ values.

To identify significantly associated genes with the MCPs per cell type, cells at the extreme ends of the MCP distribution were selected—specifically, those in the top 10% and bottom 10% of MCP scores for each cell type. The scanpy rank_genes_groups function with default parameters was subsequently used. This function conducts a $t$-test between the two cell groups to pinpoint genes that are differentially expressed, offering an adjusted $P$ value that accounts for the total number of genes assessed. We filtered for heat shock proteins to determine *HSPA1B* to be significantly differentially expressed for naive T cells (adjusted $P \leq 2.9 \times 10^{-272}$), CD8 effector memory cells (adjusted $P \leq 1.2 \times 10^{-172}$), CD4 regulatory T cells (adjusted $P \leq 5.3 \times 10^{-41}$), plasma B cells (adjusted

$P \leq 6.5 \times 10^{-34}$), CD4 central memory T cells (adjusted $P \leq 1.1 \times 10^{-1}$) and memory B cells (adjusted $P \leq 6.5 \times 10^{-37}$).

To determine if the identified genes played a role in altered cell–cell interactions, gene comparisons were made for each cell type against the NicheNet database of protein–protein interactions, using gene names as identifiers[90]. An interaction was classified as MCP associated if both the corresponding receptor and ligand were present among the significant genes (adjusted $P$ value less than 0.01) from two different cell types. An interaction was deemed MCP ligand associated if the ligand was linked to an MCP in one cell type and the receptor exhibited a normalized mean expression over 1 in another cell type. Similarly, an interaction was considered MCP receptor associated if the receptor was connected to an MCP in one cell type and the ligand had at least 10 counts in the other cell type.

### Reporting summary

Further information on research design is available in the Nature Portfolio Reporting Summary linked to this article.

## Data availability

All used datasets are available through out-of-the-box dataloaders in pertpy. We obtained the CRISPR screen dataset from Norman et al.[16], which is available in the Gene Expression Omnibus (GEO) (GSE133344). We obtained the chemical perturbation dataset from McFarland et al.[17], which the authors made available on figshare at https://figshare.com/s/139f64b495dea9d88c70. We obtained the TNBC dataset from Zhang et al.[18], which is available in the GEO with accession numbers GSE169246, GSE136206 and GSE123814.

## Code availability

The pertpy source code is available at https://github.com/scverse/pertpy under the Apache 2.0 license. Further documentation, tutorials and examples are available at https://pertpy.readthedocs.io. Scripts, notebooks and analysis results to reproduce our analysis and figures are available at https://github.com/theislab/pertpy-reproducibility.

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

## Acknowledgements

We thank all users of pertpy who regularly provide valuable feedback. We thank the differential gene expression analysis team of the scverse Hackathon in Cambridge in 2023 that developed prototypes and plots for the corresponding module in pertpy. We further thank R. K. Rubens and F. Curion for constructive comments on the paper. A.N. is supported by the Konrad Zuse School of Excellence in Learning and Intelligent Systems (ELIZA), through the DAAD program of the Konrad Zuse School of Excellence in Artificial Intelligence, sponsored by the German Federal Ministry of Education and Research. S.P. acknowledges support from Open Targets (OTAR-3083) and from an Einstein Fellow grant to C.S., with N. Blüthgen. T.D.G. and C.S. acknowledge funding from Wellcome-LEAP Delta Tissue. F.C., M.B. and K.H. are supported by the Helmholtz Association under the joint research school 'Munich School For Data Science'. K.H. acknowledges financial support from Joachim Herz Stiftung via Add-on Fellowships for Interdisciplinary Life Science.

## Author contributions

L.H. conceived of the study. L.H., Y.J., X.W., L.M., A.A.M., X.Z., A.S., J.O., E.D., M.M., F.C., I.M., S.P., T.D.G., A.T., K.H., M.D., M.B., I.G., G.S., A.N., E.R., M.L., G.P. and S.R. implemented pertpy. L.H. and L.M. analyzed the CRISPR screen dataset. L.M., L.H. and Y.J. analyzed the chemical perturbation dataset. L.M., L.H. and T.D.G. analyzed the TNBC treatment dataset. S.P., Z.Z. and L.H. performed the runtime benchmarking. L.H., L.M. and Y.J. wrote the paper. F.J.T., H.B.S. and C.S. supervised the work. All authors read, corrected and approved the final paper.

## Funding

## Competing interests

L.H. and S.R. are employees of LaminLabs. F.J.T. consults for Immunai Inc., Singularity Bio B.V., CytoReason Ltd and Omniscope Ltd and has ownership interest in Dermagnostix GmbH and Cellarity. C.S. is on the scientific advisory board of CytoReason Ltd. S.P. consults for Relation Therapeutics. T.D.G. is an employee of KiraGen Bio. G.S. is an employee of Boehringer Ingelheim International Pharma GmbH. M.L. consults for Santa Ana Bio, owns interests in Relation Therapeutics and is co-founder and equity holder at AIVIVO. The other authors declare no competing interests.

## Additional information

**Extended data** is available for this paper at https://doi.org/10.1038/s41592-025-02909-7.

**Correspondence and requests for materials** should be addressed to Fabian J. Theis.

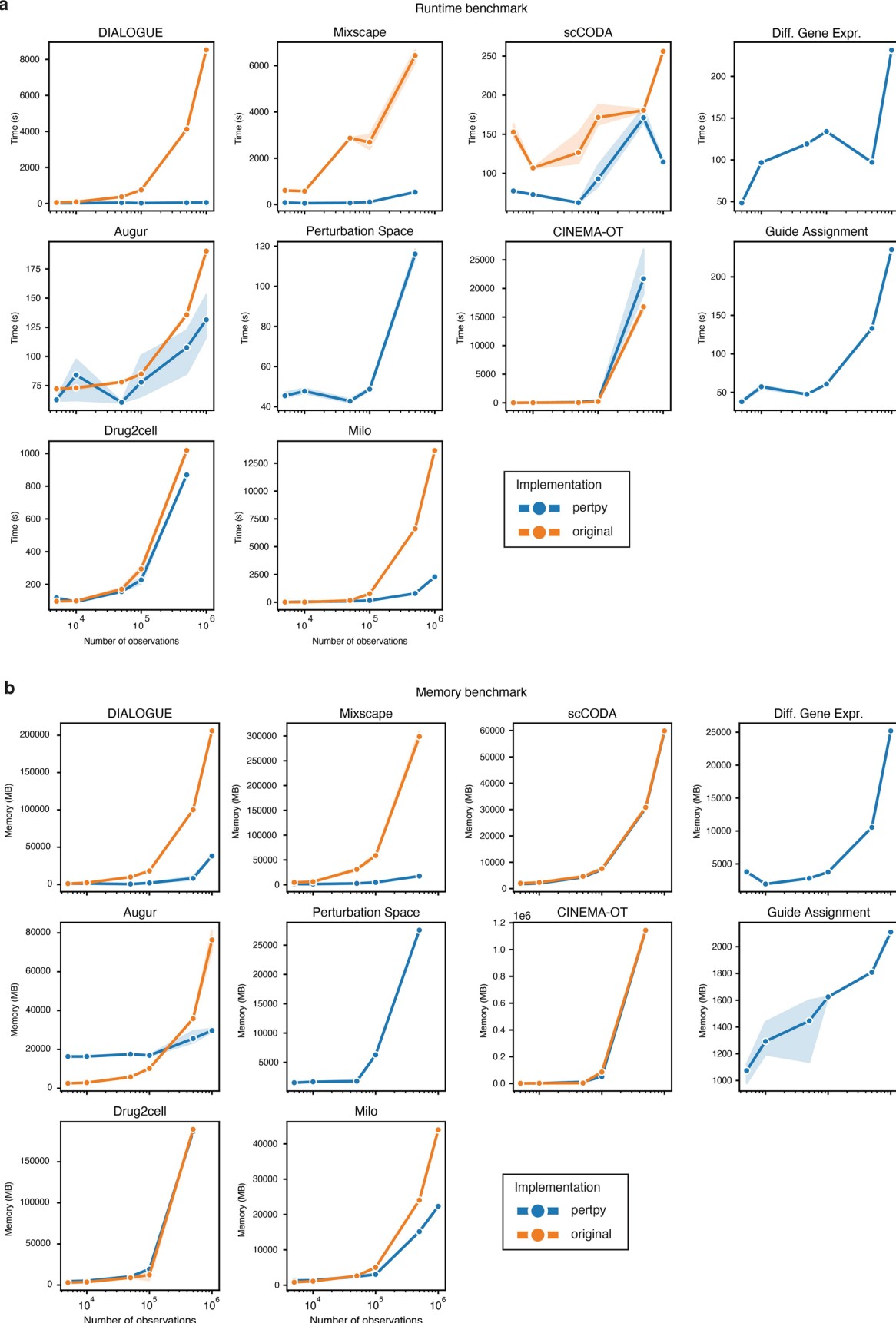

**Extended Data Fig. 1 | Runtime and memory benchmark. (a)** Runtime and (**b**) memory usage comparison of tools between pertpy's implementation and correspondingly the existing R implementation or the formerly published original implementation.

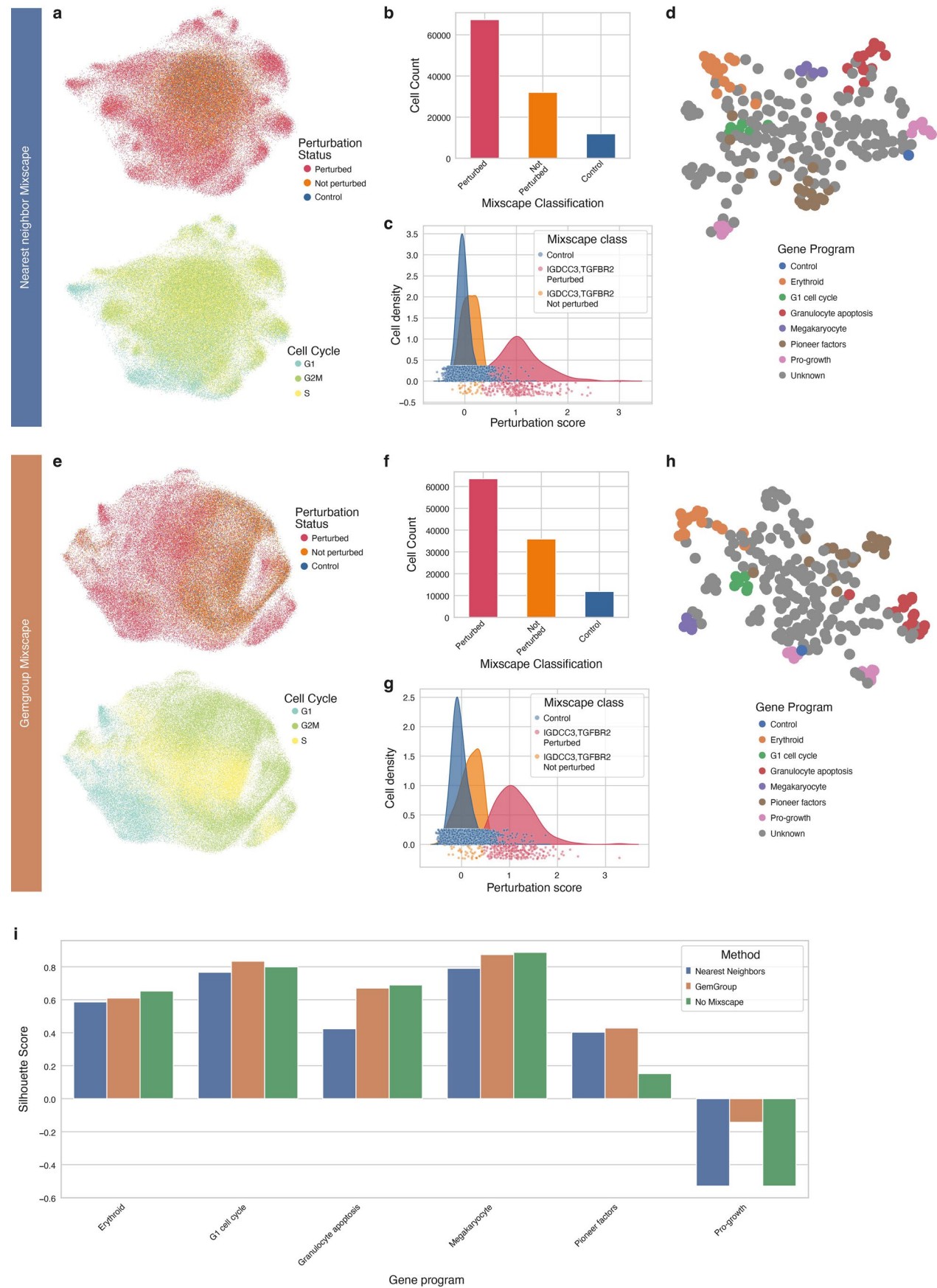

**Extended Data Fig. 2 | See next page for caption.**

**Extended Data Fig. 2 | Comparison of preprocessing strategies. (a)** UMAP representation of the perturbation signature, computed by comparing a cell's expression to its nearest neighbor control cells, thereby removing confounding factors such as cell cycle effects. (**b**) Mixscape classifies cells as successfully perturbed or targeted but not successfully perturbed. (**c**) Example perturbation score density plot for a combination gene activation. (**d**) MLPClassifier space computed after removing cells identified as not perturbed (NP). (**e**–**h**) Same as panels **a**–**d**, but for pertpy's Mixscape implementation, where the perturbation signature is computed by comparing a cell's expression to that of all control cells within the same GEM group (batch of cells processed in the same lane on a 10x Genomics chip). (**i**) Mean silhouette score per gene program for the two Mixscape preprocessing strategies shown in panels **a**–**h**, as well as for no Mixscape application (Fig. 2).

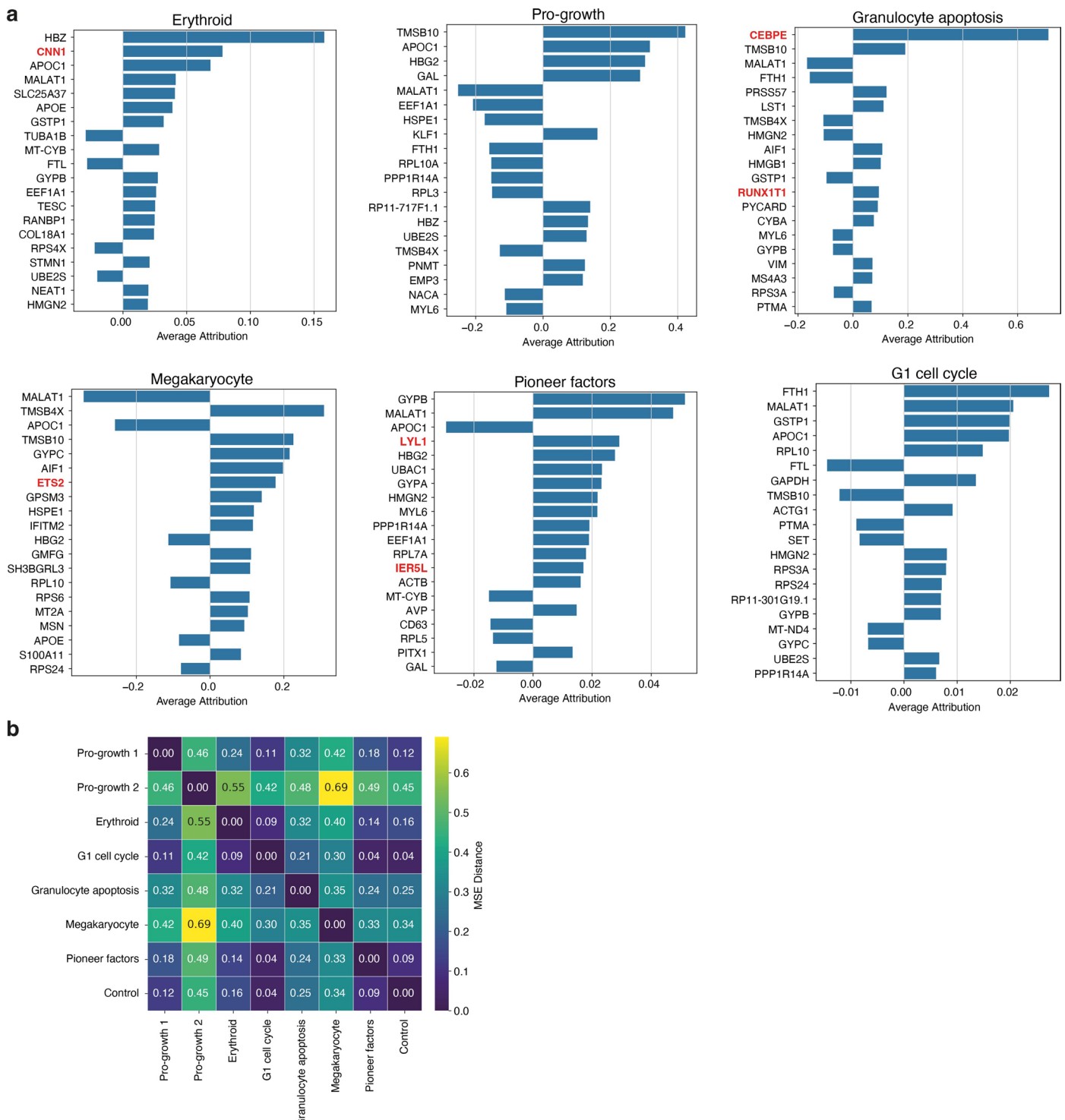

**Extended Data Fig. 3 | Integrated gradients analysis. (a)** Top 15 genes identified per gene program as most important for predicting the corresponding perturbation (single gene or gene pair) using the MLPClassifier. Gene importance was determined using integrated gradients (**Methods**). Attribution scores are shown for each gene, averaged across all cells within the respective gene program. Genes directly targeted by at least one perturbation within the gene program group are highlighted in red. **(b)** Pairwise MSE distances between gene programs in the perturbation space.

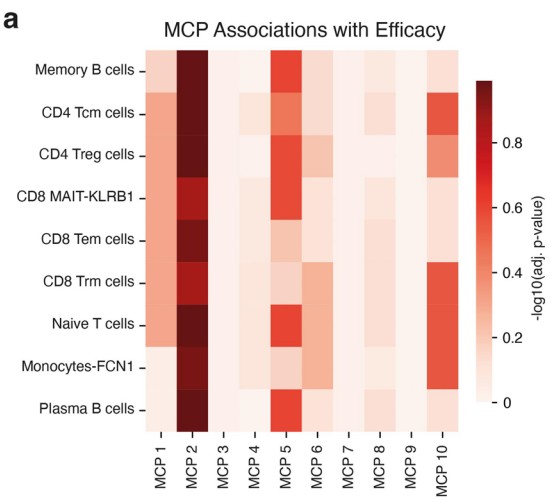

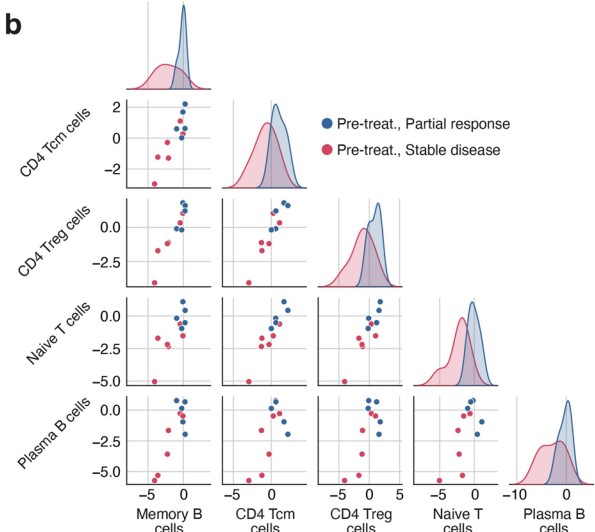

**Extended Data Fig. 4 | Multicellular programs associated with treatment response.** (**a**) DIALOGUE analysis shows several multicellular programs (MCPs) potentially associated with treatment efficacy. P-values are from independent-sample t-tests with Benjamini–Hochberg correction. Exact p-values are provided in Extended Data Table 2. (**b**) Pairplot of MCP 2. The diagonal shows a cell type specific kernel density estimate of the mean score for each MCP by sample. In the lower triangle's scatter plots, each point denotes an average patient score for the cell types labeled on the corresponding row (x-axis) and column (y-axis).

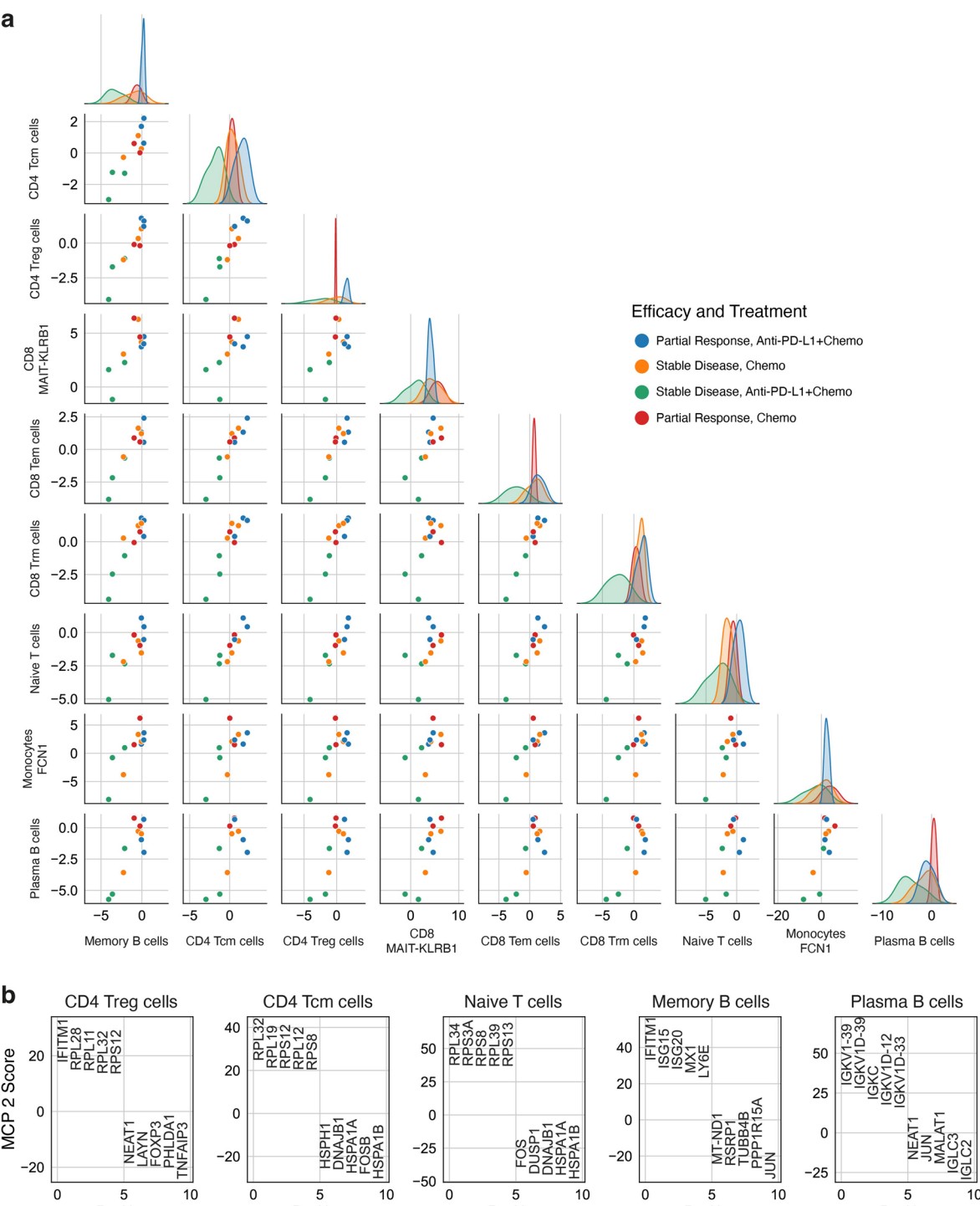

**Extended Data Fig. 5 | Cell type specificity of multicellular programs and MCP2 gene scores.** (**a**) Pair plots for MCP 2. The kernel density estimate along the diagonal shows the average score for each MCP by sample, specific to the indicated cell type. In the lower triangle's scatter plots, each point signifies the average measurement from a patient for the cell types denoted by the respective row (x-axis) and column (y-axis). MCP 2 separates poor response to the PDL-1 inhibitors. (**b**) MCP 2 extrema genes per cell type. Shown are the respective five genes with the highest and lowest scores for MCP2. HSPA1B, which is significantly increased in MCP2 for all tested cell types (**Methods**), has been previously identified as a prognostic biomarker in breast cancer[91,92].

**Extended Data Table 1 | Comparison of pertpy to other perturbation analysis frameworks**

| Toolbox | MUSIC | ScMAGeCK | SCEPTRE | GSFA | FR-Perturb | pertpy |
|---|---|---|---|---|---|---|
| Programming language | R | R | R | R | R, Python | Python |
| Ecosystem | - | Bioconductor | - | - | - | scverse |
| License | Apache-2.0 | BSD | GPL-3.0 | MIT | GPL-3.0 | MIT |
| Dataloaders | - | - | - | - | - | Yes |
| Metadata annotation | - | - | - | - | - | Yes |
| Guide assignment | - | - | Yes | - | - | Yes, e.g. Poisson-Gaussian mixture model |
| Differential gene expression | Performed, but not generalized | Performed, but not generalized | - | Performed, but not generalized | - | Yes, general interface |
| CRISPR perturbation analysis | Yes, core functionality | Yes, core functionality | Yes, core functionality | Yes, core functionality | Yes, core functionality | Yes, mixscape |
| Compositional analysis | - | - | - | - | - | Yes, scCODA 2.0, tascCODA 2.0 |
| Multicellular programs | - | - | - | Yes, latent factors | Yes, latent factors | Yes, DIALOGUE |
| Distances & metrics | Yes, topic-based | Yes, e.g. selection scores | - | - | Yes, Euclidean distance | Yes, e.g. e-distance |
| Response prediction | Yes | - | - | - | Yes | Yes, e.g. Augur |
| Perturbation embeddings | Yes, via topics | Yes, linear regression model | - | Yes, latent factors | Yes, latent factors | Yes, e.g. Pseudobulk |
| Visualizations | Yes | Yes | Yes | Yes | - | Yes |

**Extended Data Table 2 | DIALOGUE multicellular program adjusted p-values per cell type**

| | MCP 1 | MCP 2 | MCP 3 | MCP 4 | MCP 5 | MCP 6 | MCP 7 | MCP 8 | MCP 9 | MCP 10 |
|---|---|---|---|---|---|---|---|---|---|---|
| Memory B cells | 0.6829 | 0.1022 | 0.9515 | 0.9911 | 0.2550 | 0.7412 | 0.9577 | 0.8645 | 0.9990 | 0.7723 |
| CD4 central memory T cells | 0.4999 | 0.1022 | 0.9515 | 0.8197 | 0.3515 | 0.7412 | 0.9577 | 0.7628 | 0.9990 | 0.2804 |
| CD4 regulatory T cells | 0.4999 | 0.1022 | 0.9515 | 0.9761 | 0.2637 | 0.6213 | 0.9577 | 0.9589 | 0.9990 | 0.4192 |
| CD8 mucosal-associated invariant T cells | 0.4999 | 0.1392 | 0.9515 | 0.8674 | 0.2637 | 0.7880 | 0.9577 | 0.8214 | 0.9990 | 0.7723 |
| CD8 effector memory T cells | 0.4999 | 0.1115 | 0.9515 | 0.8674 | 0.6207 | 0.7880 | 0.9577 | 0.7628 | 0.9990 | 0.7723 |
| CD8 tissue-resident memory T cells | 0.4999 | 0.1392 | 0.9515 | 0.8197 | 0.6823 | 0.5370 | 0.9577 | 0.7628 | 0.9990 | 0.2804 |
| Naive T cells | 0.4999 | 0.1022 | 0.9515 | 0.8197 | 0.2550 | 0.5370 | 0.9577 | 0.7628 | 0.9990 | 0.2804 |
| Intermediate monocytes | 0.9240 | 0.1115 | 0.9515 | 0.8197 | 0.6823 | 0.5370 | 0.9577 | 0.8962 | 0.9990 | 0.2804 |
| Plasma B cells | 0.9240 | 0.1022 | 0.9515 | 0.9911 | 0.2550 | 0.7880 | 0.9577 | 0.7628 | 0.9990 | 0.7723 |

**Extended Data Table 3 | Adjusted p-values from the DIALOGUE extrema test for MCP2 across cell types and AP-1-associated genes**

| Cell Type | JUN | FOS | FOSB |
|---|---|---|---|
| CD4 regulatory T cells | 4.28 e-48 | 9.22 e-13 | 7.13 e-23 |
| CD4 central memory T cells | 6.02 e-65 | 3.94 e-66 | 1.78 e-117 |
| CD8 effector memory T cells | 4.71 e-119 | 5.57 e-115 | 3.10 e-186 |
| Intermediate monocytes | 0.025122 | 0.827241 | 8.0 e-6 |
| Memory B cells | 1.54 e-193 | 5.39 e-98 | 1.17 e-142 |
| Naive T cells | 2.15 e-297 | 1.14 e-221 | 1.49 e-185 |
| Plasma B cells | 1.05 e-26 | 1.57 e-34 | 1.18 e-17 |

**Extended Data Table 3 | Adjusted p-values from the DIALOGUE extrema test for MCP2 across cell types and AP-1-associated genes**

# Reporting Summary

## Statistics

For all statistical analyses, confirm that the following items are present in the figure legend, table legend, main text, or Methods section.

| n/a | Confirmed | |
|---|---|---|
| ☐ | ☒ | The exact sample size (*n*) for each experimental group/condition, given as a discrete number and unit of measurement |
| ☒ | ☐ | A statement on whether measurements were taken from distinct samples or whether the same sample was measured repeatedly |
| ☐ | ☒ | The statistical test(s) used AND whether they are one- or two-sided *Only common tests should be described solely by name; describe more complex techniques in the Methods section.* |
| ☐ | ☒ | A description of all covariates tested |
| ☐ | ☒ | A description of any assumptions or corrections, such as tests of normality and adjustment for multiple comparisons |
| ☐ | ☒ | A full description of the statistical parameters including central tendency (e.g. means) or other basic estimates (e.g. regression coefficient) AND variation (e.g. standard deviation) or associated estimates of uncertainty (e.g. confidence intervals) |
| ☐ | ☒ | For null hypothesis testing, the test statistic (e.g. *F*, *t*, *r*) with confidence intervals, effect sizes, degrees of freedom and *P* value noted *Give P values as exact values whenever suitable.* |
| ☒ | ☐ | For Bayesian analysis, information on the choice of priors and Markov chain Monte Carlo settings |
| ☒ | ☐ | For hierarchical and complex designs, identification of the appropriate level for tests and full reporting of outcomes |
| ☒ | ☐ | Estimates of effect sizes (e.g. Cohen's *d*, Pearson's *r*), indicating how they were calculated |

*Our web collection on statistics for biologists contains articles on many of the points above.*

## Software and code

Policy information about availability of computer code

| Data collection | All used datasets are available through out-of-the-box dataloaders in pertpy. We obtained the CRISPR screen dataset from Norman et al.20 which is available in GEO (GSE133344). We obtained the chemical perturbation dataset from McFarland et al.21 which the authors made available on figshare at https://figshare.com/s/139f64b495dea9d88c70. We obtained the TNBC dataset from Zhang et al.22 which is available on GEO with accession numbers GSE169246, GSE136206, and GSE123814. |
|---|---|
| Data analysis | The main analysis was performed with our Python software pertpy (version 0.11.2) available at https://github.com/scverse/pertpy. |

For manuscripts utilizing custom algorithms or software that are central to the research but not yet described in published literature, software must be made available to editors and reviewers. We strongly encourage code deposition in a community repository (e.g. GitHub). See the Nature Portfolio guidelines for submitting code & software for further information.

## Data

Policy information about availability of data

All manuscripts must include a data availability statement. This statement should provide the following information, where applicable:

- Accession codes, unique identifiers, or web links for publicly available datasets
- A description of any restrictions on data availability
- For clinical datasets or third party data, please ensure that the statement adheres to our policy

All used datasets are available through out-of-the-box dataloaders in pertpy. We obtained the CRISPR screen dataset from Norman et al.20 which is available in GEO (GSE133344). We obtained the chemical perturbation dataset from McFarland et al.21 which the authors made available on figshare at https://figshare.com/s/139f64b495dea9d88c70. We obtained the TNBC dataset from Zhang et al.22 which is available on GEO with accession numbers GSE169246, GSE136206, and GSE123814.

## Research involving human participants, their data, or biological material

Policy information about studies with human participants or human data. See also policy information about sex, gender (identity/presentation), and sexual orientation and race, ethnicity and racism.

| Reporting on sex and gender | NA |
|---|---|
| Reporting on race, ethnicity, or other socially relevant groupings | NA |
| Population characteristics | NA |
| Recruitment | NA |
| Ethics oversight | NA |

Note that full information on the approval of the study protocol must also be provided in the manuscript.

# Field-specific reporting

Please select the one below that is the best fit for your research. If you are not sure, read the appropriate sections before making your selection.

☒ Life sciences      ☐ Behavioural & social sciences      ☐ Ecological, evolutionary & environmental sciences

For a reference copy of the document with all sections, see nature.com/documents/nr-reporting-summary-flat.pdf

# Life sciences study design

All studies must disclose on these points even when the disclosure is negative.

| Sample size | Sample size was determined by the previously published datasets that were used for this study. Datasets were chosen in order to describe the functionality of the software. |
|---|---|
| Data exclusions | The Method section of the respective analyses notes data exclusions due to various quality control and feature selection steps. |
| Replication | All attempts to computationally replicate the data analysis were successful. No replication of experimental data was performed since we did not collect any experimental data. |
| Randomization | Randomization is not applicable as we did not collect any experimental data. |
| Blinding | Blinding was not applicable as we solely performed computational analyses where dataset and context matters. |

# Reporting for specific materials, systems and methods

We require information from authors about some types of materials, experimental systems and methods used in many studies. Here, indicate whether each material, system or method listed is relevant to your study. If you are not sure if a list item applies to your research, read the appropriate section before selecting a response.

## Materials & experimental systems

| n/a | Involved in the study |
|---|---|
| ☒ ☐ | Antibodies |
| ☒ ☐ | Eukaryotic cell lines |
| ☒ ☐ | Palaeontology and archaeology |
| ☒ ☐ | Animals and other organisms |
| ☒ ☐ | Clinical data |
| ☒ ☐ | Dual use research of concern |
| ☒ ☐ | Plants |

## Methods

| n/a | Involved in the study |
|---|---|
| ☒ ☐ | ChIP-seq |
| ☒ ☐ | Flow cytometry |
| ☒ ☐ | MRI-based neuroimaging |

## Plants

| | |
|---|---|
| Seed stocks | *Report on the source of all seed stocks or other plant material used. If applicable, state the seed stock centre and catalogue number. If plant specimens were collected from the field, describe the collection location, date and sampling procedures.* |
| Novel plant genotypes | *Describe the methods by which all novel plant genotypes were produced. This includes those generated by transgenic approaches, gene editing, chemical/radiation-based mutagenesis and hybridization. For transgenic lines, describe the transformation method, the number of independent lines analyzed and the generation upon which experiments were performed. For gene-edited lines, describe the editor used, the endogenous sequence targeted for editing, the targeting guide RNA sequence (if applicable) and how the editor was applied.* |
| Authentication | *Describe any authentication procedures for each seed stock used or novel genotype generated. Describe any experiments used to assess the effect of a mutation and, where applicable, how potential secondary effects (e.g. second site T-DNA insertions, mosiacism, off-target gene editing) were examined.* |

