## [Peer Review File · Nature Methods]

Pertpy: an end-to-end framework for perturbation analysis

Corresponding Author: Professor Fabian Theis

Version 0:

Decision Letter:

10th Sep 2024

Dear Professor Theis,

Your Brief Communication, "Pertpy: an end-to-end framework for perturbation analysis", has now been seen by 2 reviewers. As you will see from their comments below, although the reviewers find your work of considerable potential interest, they have raised a number of concerns. We are interested in the possibility of publishing your paper in Nature Methods, but would like to consider your response to these concerns before we reach a final decision on publication.

We therefore invite you to revise your manuscript to fully address all these concerns.

Link Redacted

We hope to receive your revised paper within 4 months. If you cannot send it within this time, please let us know. In this event, we will still be happy to reconsider your paper at a later date so long as nothing similar has been accepted for publication at Nature Methods or published elsewhere.

OPEN SCIENCE REQUIREMENTS

REPORTING SUMMARY AND EDITORIAL POLICY CHECKLISTS

DATA AVAILABILITY

All novel DNA and RNA sequencing data, protein sequences, genetic polymorphisms, linked genotype and phenotype data, gene expression data, macromolecular structures, and proteomics data must be deposited in a publicly accessible database, and accession codes and associated hyperlinks must be provided in the "Data Availability" section.

CODE AVAILABILITY

Please include a "Code Availability" subsection in the Online Methods which details how your custom code is made available. Only in rare cases (where code is not central to the main conclusions of the paper) is the statement "available upon request" allowed (and reasons should be specified).

For more information on our code sharing policy and requirements, please see: <https://www.nature.com/nature-research/editorial-policies/reporting-standards#availability-of-computer-code>

MATERIALS AVAILABILITY

ORCID

Nature Methods is committed to improving transparency in authorship. As part of our efforts in this direction, we are now requesting that all authors identified as 'corresponding author' on published papers create and link their Open Researcher and Contributor Identifier (ORCID) with their account on the Manuscript Tracking System (MTS), prior to acceptance. This applies to primary research papers only. ORCID helps the scientific community achieve unambiguous attribution of all scholarly contributions. You can create and link your ORCID from the home page of the MTS by clicking on 'Modify my Springer Nature

account'. For more information please visit www.springernature.com/orcid.

Sincerely,

Lin Tang, PhD
Senior Editor
Nature Methods

Reviewers' Comments:

Reviewer #1 (Remarks to the Author):

First of all, this package obviously addresses a pressing need. As single-cell functional genomics methods have taken off, computational analysis remains a major roadblock for uptake. This paper and the associated python package `pertpy` describe an extension to the ubiquitous `scanpy` framework for analyzing single-cell RNA sequencing that focuses on analyzing various types of single-cell perturbation experiments. The authors have a superb track record of producing tools that are well engineered, well maintained, and widely used by the community. The package also balances reimplementing old algorithms, with the goal of making them more robust and more widely available, with some innovation. I think it's overall an extremely useful contribution that should be published. In fact, given the likelihood of wide uptake, my primary reservations focus on ensuring that the implementations and analyses presented are sound, as I think they are likely to influence how many labs will analyze experimental results in the future. With these concerns addressed the paper would be suitable for publication in Nature Methods.

Thomas Norman

Major comments:

1) `pertpy` contains faster reimplementations of several algorithms, including `mixscape`, `augur`, `scCODA`, and `DIALOGUE`. This is great for the community. However, I think it would be good to include sanity checks showing that these reimplementations yield results that are substantially similar to the parent tool on particular datasets. Otherwise we run the risk of having identically named tools that produce different results. For example, in the section on MIX-Seq it states that it allows replication and extension of the original analyses, but focuses on a set of new results. Are the original trametinib results replicated in at least a broad sense? When I looked through the notebooks for `DIALOGUE` analysis it seemed as though the different solvers potentially gave different results even within `pertpy`.

2) To summarize a series of more detailed comments I will separate into their own section below: for Perturb-seq experiments, I wish the results presented an alternative workflow that did less processing than what is currently included. Though I understand the desire/enthusiasm to show off `pertpy`'s capabilities, I am wary that the workflow included in the paper will instead be taken as establishing best practices. I understand the desire to not do analyses of five different datasets to demonstrate different capabilities, but I think it would be helpful to present the results alongside a simpler workflow to make clear the benefit of the heavier processing steps.

3) Figure 4E looks strange. None of the programs are significantly enriched according to Supplementary Table 1 and there seems to be a strong correlation across different cell types in the p-values. (Side note, the choice of the $1.1e-1$ p-value threshold for significance is peculiar.) Perhaps I am missing the point, but what exactly are these programs meant to represent if they are not enriched in any particular subpopulation?

4) The connections between MCP2 and AP-1 components are not clear from the text, though I see some of the gene names in Supplementary Figure 2. Are these genes in the MCP2 program? Can you please add the gene programs as a supplementary table? I am also confused by the need for the new procedure based on `rank_genes_groups` to identify cell-type-specific genes in the MCPs. I thought this was what `DIALOGUE` produces as output?

Detailed comments about Perturb-seq: Please note that I offer these comments in the spirit of being constructive about how to best analyze these datasets, for which there is obviously no universally correct solution.

* Removing "confounders" via local perturbation signatures: I have reservations about this process. In practice this amounts to subtracting an average of 20 nearby control cells chosen based on a somewhat arbitrary nearest neighbor graph. To me this seems likely to introduce noise, particularly for lowly expressed genes and for perturbations that cause major departures from the control state, where the nearest neighbor calculation is fraught. It also presumes that cells with a perturbation do not exhibit compositional differences with respect to variation seen within the control cells. For example, you largely remove the effect of the cell cycle, as you see in your plots. It's not clear to me that this is a good thing to do as I don't view the cell cycle as a "confounder." (For example, I would view which `gemgroup` cells came from as a true confounder.) In particular, several of the

perturbations in this experiment target core cell cycle regulators, so you are potentially dampening their effects. (E.g., for a perturbation that caused all cells to perfectly accumulate in G1, this local, nonlinear offsetting procedure would subtract the effect of the perturbation entirely, on average.) I am wondering if this is what is driving the TP73 result: are you normalizing out the effects of the CDKN2A perturbations, causing them to cluster with things that they didn't cluster with before? In general, in our work we have always favored the simpler procedure of comparing to control cells in the same gemgroup, which still removes technical batch effects but has the advantage of being independent of choices about expression normalization, metrics used for computing nearest neighbors, etc. I question how reproducible the "nearest" control cells are across different choices.

* Removing "unperturbed" cells: Firstly, this dataset is a CRISPRa dataset, not CRISPR knockout. Thus references in the text and figures to "KO" or "knockout" should be modified. The underlying assumption in the Mixscape model is that there are cells that have escaped perturbation. For CRISPR cutting, there is a mechanistic expectation that this occurs because cutting sometimes leaves in-frame indels. For other modalities like CRISPRi/CRISPRa it is less clear what these "unperturbed" cells mean. It is fine to remove them as a phenomenological choice in analysis, but I think the norm in the field should be to avoid doing so unless it's clearly necessary. If you keep these cells in do any of your results actually change?

* Label transfer: I get the idea of demonstrating this capability, but I am not sure these are the right annotations to expand since they aren't ground truth. The clusters underlying the annotations were defined using HDBSCAN on the expression profiles. HDBSCAN is intrinsically conservative and is meant to identify clusters that are robust. By extension, the expansions defined by label transfer are not robust. If you keep applying label transfer to your new clusters they will just expand until they encompass the whole dataset, right? Similarly, a new cluster is certainly possible, but that just means that it wasn't robust enough to be selected by HDBSCAN from the raw expression data. Which perturbations fall into that cluster?

* Perturbation spaces: The "perturbation space" presented amounts to another way of embedding mean profiles, in this case based on a classifier. There is no one correct way to do embeddings, so I'm not sure what to make of the claims about clusters splitting etc., which appear to be being made based on the visualization rather than any mathematical criterion. Indeed, this dataset is treated separately in the (very helpful) pertpy tutorials online where many of the different embeddings appear to have similar structure to the original paper's embedding and preserve the cluster structure observed there. That suggests that it's the preprocessing steps that are causing the differences that are emphasized in this paper. (As noted above, I suspect it's the local perturbation signatures.) It also suggests some of the new claims may be brittle to choices about what goes into producing the embedding. One thing I would check is whether the classifier is picking up the CRISPRa target gene activation (e.g. of KLF1), and then using that to split perturbations in the embedding. I think it would be good to present the analysis with and without different choices of preprocessing steps to show how they influence the results. This could be used to demonstrate how pertpy can enable rapid testing of analytical pipelines.

Minor comments:

* Can you include a version of Figure 2A colored by gemgroup? It's a common source of batch effects.

* Energy statistics were introduced for perturbation testing in <https://doi.org/10.1016/j.cell.2022.05.013>. This is referenced in the scPerturb paper, which is instead cited as the source here.

* "To rank perturbation effects, we used pertpy to calculate the Euclidean distance between the pre- and post-treatment patients of the four groups due to its superior performance in independent benchmarks" Perhaps "strong performance" is better phrasing here? The cited paper shows several distances perform similarly.

* In Fig 4B the UMAP axes are meaningless and do not need to be included/labeled.

* There are a few topics that are pretty basic that I think could be included if there is room, though I understand if the authors do not have time. For example, differential expression analysis is one of the most popular analytical tasks but it is only mentioned in passing. Guide calling is also given little coverage even though this is one of the more challenging parts of these experiments. Finally, I'm a bit surprised that there was no discussion of batch effects or batch correction. For functional genomics experiments this is a common problem since they are usually large enough that they have to be done in batched form.

Reviewer #1 (Remarks on code availability):

We use the code in my lab already in some instances, and the authors have an extremely good track record of producing widely used tools. As noted in my review, I do think one thing that is missing and unclear is how these python reimplementations compare to the original packages.

Reviewer #2 (Remarks to the Author):

See attached.

Version 1:

Decision Letter:

Our ref: NMETH-A57419A

21st Jun 2025

Dear Dr. Theis,

Thank you for submitting your revised manuscript "Pertpy: an end-to-end framework for perturbation analysis" (N METH-A57419A). It has now been seen by the original referees and their comments are below. The reviewers find that the paper has improved in revision, and therefore we'll be happy in principle to publish it in Nature Methods, pending minor revisions to satisfy the referees' final requests and to comply with our editorial and formatting guidelines.

Given the current length and content of this paper, we think Article would be a more suitable manuscript type for it and have changed its manuscript type (and manuscript #) accordingly.

TRANSPARENT PEER REVIEW

ORCID

Sincerely,

Lin Tang, PhD
Senior Editor
Nature Methods

Reviewer #1 (Remarks to the Author):

The authors have gone above and beyond in responding to my comments. I am particularly pleased that they now treat some other common analyses such as guide calling, differential expression, and batch correction. These were purely optional suggestions. The package is clearly of great value for the community, in line with the authors' previous work, and is already in use in my lab. I enthusiastically recommend publication.

Reviewer #1 (Remarks on code availability):

I think the authors set the standard for high-quality tools and documentation.

Reviewer #2 (Remarks to the Author):

The authors have addressed my primary concerns. There is an issue with their code, which I mention below in "Remarks on code availability".

Reviewer #2 (Remarks on code availability):

The code at <https://github.com/theislab/pertpy-reproducibility/tree/main/benchmark> does not seem to actually carry out the analyses underlying Supplementary Figure 1.

Version 2:

Decision Letter:

16th Oct 2025

Dear Professor Theis,

I am pleased to inform you that your Article, "Pertpy: an end-to-end framework for perturbation analysis", has now been accepted for publication in Nature Methods. The received and accepted dates will be 4th Aug 2024 and 16th Oct 2025. This note is intended to let you know what to expect from us over the next month or so, and to let you know where to address any further questions.

Over the next few weeks, your paper will be copyedited to ensure that it conforms to Nature Methods style. Once your paper is typeset, you will receive an email with a link to choose the appropriate publishing options for your paper and our Author Services team will be in touch regarding any additional information that may be required. It is extremely important that you let us know now whether you will be difficult to contact over the next month. If this is the case, we ask that you send us the contact information (email, phone and fax) of someone who will be able to check the proofs and deal with any last-minute problems.

Authors may need to take specific actions to achieve compliance with funder and institutional open access mandates.

If your research is supported by a funder that requires immediate open access (e.g. according to [Plan S principles](https://www.springernature.com/gp/open-science/plan-s-compliance) or the [NIH public access policy](https://www.springernature.com/gp/open-science/us-federal-agency-compliance)) then you should select the gold OA route, and we will direct you to the compliant route where possible. Because authors warrant under our subscription licensing terms that they haven't committed to licensing any version of their article under a licence inconsistent with the terms of our agreement – including the applicable embargo period – publication under the subscription model isn't suitable for authors whose funders require no embargo.

Please feel free to contact me if you have questions about any of these points. Thank you very much for publishing your paper with Nature Methods!

Best regards,

Lin Tang, PhD
Senior Editor
Nature Methods

** Visit the Springer Nature Editorial and Publishing website at www.springernature.com/editorial-and-publishing-jobs for more information about our career opportunities. If you have any questions please click here.**

Response to reviewers: Pertpy: an end-to-end framework for perturbation analysis - NMETH-BC57419

Dear Lin,

Thank you very much for handling our manuscript and providing comments & reviews. We were very happy with the positive and supportive reviewers' comments and constructive (but quite work-intense) suggestions for the revision.

Based on the comments, we have identified these key issues with the initial submission: first, it was missing proof that our implementations are equal to the existing implementations and scale to large dataset sizes; second, it lacked of comparison of preprocessing strategies for the CRISPRa dataset from Thomas Norman; and finally, the multicellular program analysis using DIALOGUE was difficult to understand.

To address these concerns we have made the following major changes:

- We show that our implementations closely align with existing methods, perform comparably or better in speed, and scale effectively in memory usage to large datasets.
- We overhauled and simplified the use case of the CRISPRa dataset of Thomas Norman, now focusing on pertpy's ability to quickly iterate with different preprocessing methods.
- We refined how we describe the exploratory multicellular program analysis with DIALOGUE by clarifying the interpretation of p-values, detailing the criteria for identifying MCP2-associated genes, and explaining how our ligand–receptor analysis supports this approach. We also added several supplementary tables to allow readers to better interpret our findings.

During the revisions, we further improved all tools of pertpy with respect to speed, usability, and tutorials. Additionally, we implemented a new Poisson-Gaussian mixture model for guide assignment, introduced an entropy-based score to assess uncertainty when using label transfer, and extended the metadata annotation module to support querying drug viability data from PRISM. Altogether we are convinced that we have not only fully addressed the reviewers' comments but also provided a more useful pertpy toolbox to the community, and look forward to your feedback.

In the following, we present our response to the reviewers' comments. We include **reviewer comments (black)**, **point-by-point responses (green)** and, in parts, copy **original**

manuscript text (blue) and ***modified manuscript text (blue, italic)*** or specific figure panels into our responses.

Regards,
Fabian Theis

Editor comments

Dear Professor Theis,

Your Brief Communication, "Pertpy: an end-to-end framework for perturbation analysis", has now been seen by 2 reviewers. As you will see from their comments below, although the reviewers find your work of considerable potential interest, they have raised a number of concerns. We are interested in the possibility of publishing your paper in Nature Methods, but would like to consider your response to these concerns before we reach a final decision on publication.

We therefore invite you to revise your manuscript to fully address all these concerns.

Thank you again for the comments and reviews, we outline our answers below.

Reviewer 1

First of all, this package obviously addresses a pressing need. As single-cell functional genomics methods have taken off, computational analysis remains a major roadblock for uptake. This paper and the associated python package `pertpy` describe an extension to the ubiquitous `scanpy` framework for analyzing single-cell RNA sequencing that focuses on analyzing various types of single-cell perturbation experiments. The authors have a superb track record of producing tools that are well engineered, well maintained, and widely used by the community. The package also balances reimplementing old algorithms, with the goal of making them more robust and more widely available, with some innovation. I think it's overall an extremely useful contribution that should be published. In fact, given the likelihood of wide uptake, my primary reservations focus on ensuring that the implementations and analyses presented are sound, as I think they are likely to influence how many labs will analyze experimental results in the future. With these concerns addressed the paper would be suitable for publication in *Nature Methods*.

Thomas Norman

We appreciate the reviewer's thorough and thoughtful suggestions, and are thrilled to hear that our software is already in use by their lab. Thanks for these kind and important changes, which we implemented as outlined below.

Major comments

1. `pertpy` contains faster reimplementations of several algorithms, including `mixscape`, `augur`, `scCODA`, and `DIALOGUE`. This is great for the community. However, I think it would be good to include sanity checks showing that these reimplementations yield results that are substantially similar to the parent tool on particular datasets. Otherwise we run the risk of having identically named tools that produce different results.

We strongly agree with the reviewer that our new and maintained implementations should yield substantially similar results to the existing (often unmaintained) solutions. We therefore compared all of our implementations to the existing implementations with respect to correctness, speed and memory usage. Moreover, we were in contact with the original authors of `DIALOGUE`, `scCODA`, `tascCODA`, and `Augur` during the implementation process to ensure that the implementations are correct. The implementations of `pertpy` and the existing (unmaintained) versions will never match exactly due to inherent numerical differences across programming languages, variations in core numerical libraries like `NumPy` and `JAX`, and discrepancies in preprocessing steps such as PCA. We added all of our findings in the corresponding *Methods* section of the implementations:

Concerning `DIALOGUE`:

Notably, users should be aware that the `Seurat` and `scanpy` implementations calculate PCA differently, resulting in downstream differences in MCP scores. When the same PCA

representation is used, the MCP values between the R and Python implementation have an average of 0.96 Pearson correlation when tested on the sample dataset provided in the R tutorial.

Concerning Mixscape:

The implementation was verified by comparing the classification results between the original Seurat Mixscape implementation and the pertpy implementation through a confusion matrix, showing high agreement with 4674 KO, 13098 NP, and 2386 NT cells correctly classified by both implementations, with only minor disagreements (438 cells classified as NP by pertpy but KO by original, and 133 cells classified as KO by pertpy but NP by original). Additionally, the perturbation signature scores between implementations show a strong correlation of 0.97 (p -value < 0.0001), confirming that pertpy's implementation closely reproduces the original method's quantitative measurements.

Concerning Augur:

The implementation was verified by comparing the results from pertpy's Augur implementation and the original R-based Augur package, showing excellent agreement in both default and velocity mode. The AUC scores from both implementations were highly consistent across all tested cell types with all data points falling within 4% of the expected $y=x$ line. This close correspondence was observed in both analysis modes, confirming that pertpy's implementation faithfully reproduces the computational methodology of the original R package.

Concerning scCODA:

The implementation was verified by comparing parameter estimates and log₂-fold changes with the original implementation across multiple test scenarios, including different reference cell types and treatment conditions, with results showing nearly identical values between implementations (within ~ 0.01 for parameters and ~ 0.005 for log₂-fold changes).

Concerning Milo:

The implementation was verified by comparing the results from the pertpy implementation and the original miloR package, showing a strong correlation ($r = 0.987$) between log₂FC values calculated at the cell level. Additionally, precision and recall analysis across different significance thresholds demonstrated high concordance between the two implementations, with both metrics approaching 1.0 as the threshold increases. This confirms that pertpy's Milo implementation accurately reproduces the statistical findings of the original method.

Concerning CinemaOT:

The implementation was verified by comparing the results from pertpy's CINEMA-OT implementation and the original CINEMA-OT package. Tests showed strong agreement between both implementations with a relative Frobenius norm difference of less than 0.1 (0.0973) for the optimal transport transformed confounders. Additionally, single-cell treatment effects showed exceptionally high correlation between implementations, with

mean Pearson correlation of 0.989 and mean Spearman correlation of 0.983 across all genes. Both implementations consistently revealed the same biological insight regarding distinct treatment effects in monocytes, confirming that pertpy's implementation faithfully reproduces the computational methods of the original tool.

Concerning Drug2cell:

The implementation was verified by comparing the results from pertpy's enrichment module and the original drug2cell package, demonstrating exact equivalence in both overrepresentation and enrichment analyses. Tests confirmed that the pertpy implementation produces identical results for hypergeometric overrepresentation testing in cell type-specific pathways and gene set enrichment analysis (GSEA), with all results being equal.

All scripts and notebooks to benchmark and evaluate our implementations are publicly available on <https://github.com/theislab/pertpy-reproducibility>.

For example, in the section on MIX-Seq it states that it allows replication and extension of the original analyses, but focuses on a set of new results. Are the original trametinib results replicated in at least a broad sense?

Good point. Our original goal was not to precisely reproduce the analysis in McFarland et al. but rather to demonstrate that pertpy's metadata annotation module (in this case, for GDSC scores) can facilitate various downstream analyses, such as those presented in the McFarland paper. However, we acknowledge the importance of showing that this approach yields similar results to the original study.

Hence, we have now extended pertpy's metadata annotation to enable an analysis more closely aligned with the original analysis. Specifically, we extended the metadata module to include PRISM viability data in addition to the existing GDSC drug viability data. Furthermore, we now query the GDSC AUC in addition to the $\ln(\text{IC}_{50})$ to enable direct comparability with the PRISM AUC data.

We adapted the analysis for the drugs dabrafenib and trametinib, as presented in the original MIX-seq paper, by taking the mean rank-normalized PRISM, GDSC 1, and GDSC 2 AUC values per drug. For dabrafenib, this adjustment results in more cell lines with annotated cell viability, and the findings align with those shown in the manuscript. For trametinib, our findings broadly align with those of the original paper. For example, the two genes highlighted in the study (EGR1 as a viability-independent gene and RRM2 as a highly viability-dependent gene) are also identified as such in our analysis.

The notebook presenting these results can be found here:

https://github.com/theislab/pertpy-reproducibility/blob/main/mcfarland/04_mcfarland_drug_response.ipynb

When I looked through the notebooks for DIALOGUE analysis it seemed as though the different solvers potentially gave different results even within pertpy.

The reviewer is correct in observing that the DIALOGUE solvers give different results. As we note in the manuscript, we now provide an additional solver for the MultiCCA optimization

within DIALOGUE in addition to the original solver. This new linear programming solver, unlike the original solver, is input-order invariant: it returns the same answer regardless of the order in which cell types are provided. We consulted the first author of DIALOGUE and the order dependency was not an intended effect (<https://github.com/livnatje/DIALOGUE/issues/29>) - as one would expect. As cell types do not biologically have an “order”, we have included the linear programming solver for users who may want to compute results without worrying about cell type order, particularly across multiple datasets. For users for whom order invariance is not a concern, the original solver and original parameters remain the default parameters and as shown above, we ensured that our DIALOGUE implementation matches the original R implementation given the same solver and order of cell types.

1. To summarize a series of more detailed comments I will separate into their own section below: for Perturb-seq experiments, I wish the results presented an alternative workflow that did less processing than what is currently included. Though I understand the desire/enthusiasm to show off pertpy's capabilities, I am wary that the workflow included in the paper will instead be taken as establishing best practices. I understand the desire to not do analyses of five different datasets to demonstrate different capabilities, but I think it would be helpful to present the results alongside a simpler workflow to make clear the benefit of the heavier processing steps.

We appreciate and agree with the reviewer's concern about setting appropriate methodological standards. While we initially aimed to showcase pertpy's diverse and advanced capabilities, we recognize this could inadvertently establish overly complex workflows as best practices. In response, we have streamlined our analysis examples in the manuscript and our tutorials. For the Perturb-seq use case specifically, we now focus more strongly on comparing preprocessing approaches, allowing users to better understand when simpler methods may suffice versus when more sophisticated processing steps add meaningful value. We refer to our following answers for explicit changes.

2. Figure 4E looks strange. None of the programs are significantly enriched according to Supplementary Table 1 and there seems to be a strong correlation across different cell types in the p-values. (Side note, the choice of the $1.1e-1$ p-value threshold for significance is peculiar.) Perhaps I am missing the point, but what exactly are these programs meant to represent if they are not enriched in any particular subpopulation?

The correlation across different cell types in the p-values is due to the fact that the programs are explicitly calculated using patient averages and designed to pick up differences in cell types that are shared across patients. Thus, in general, we would expect that if the value of an MCP in one cell type correlated with treatment response in the patient population, the value of that MCP in a different cell type would show a similar correlation. The listed p-value is reporting the smallest p-values across all cell types. We would like to refer the reviewer to **Supplementary Table 2** where they are now listed. The programs are best thought of as meaningful representations of patients rather than of individual cells.

Nevertheless, we agree that this analysis is hard to follow and the application of DIALOGUE to this dataset falls in the realm of “can do”. We therefore substantially shortened this section and softened the conclusions concerning the takeaways:

We applied pertpy's implementation of DIALOGUE⁴⁵, which finds MCPs using matrix decomposition in conjunction with a novel, fast input-order invariant linear programming solver, to the TNBC treatment dataset to uncover 10 multicell type signatures that may be associated with treatment response (**Methods**). *Exploratory analysis of average MCP2 scores across seven distinct cell types in each patient (Supplementary Table 2) indicated a potential association with treatment response for both treatment groups, based on cell type-specific t-tests (adj. P ≤ 1.1e-01) (Supplementary Figure 3A-B, Supplementary Figure 4A,B). Initial investigations of the MCP2-associated genes suggest involvement in heat shock protein activity and cytokine signaling (Methods, Supplementary Figure 4, Supplementary Materials), including an interaction between IL-7 and its receptor IL7R in T cells, which are known to have an antitumor role across diverse cancers⁶⁷. Increased IL-7 activity may contribute to suboptimal treatment outcomes by affecting T cell behavior and elevating levels of MCP2 associated genes JUN, FOS, and FOSB (Supplementary Table 3), which are key components of the AP-1 complex that can either inhibit or promote tumor growth, depending on the context^{68,69}.*

4.

4.1 The connections between MCP2 and AP-1 components are not clear from the text, though I see some of the gene names in Supplementary Figure 2. Are these genes in the MCP2 program? Can you please add the gene programs as a supplementary table?

Yes, the mentioned genes JUN, FOS, and FOSB, which are key components of the AP-1 complex, are statistically significantly associated with the MCP2 program. We added **Supplementary Table 3** to the manuscript which shows all of the obtained p-values.

Cell Type	JUN	FOS	FOSB
CD4 regulatory T cells	4.28 e-48	9.22 e-13	7.13 e-23
CD4 central memory T cells	6.02 e-65	3.94 e-66	1.78 e-117
CD8 effector memory T cells	4.71 e-119	5.57 e-115	3.10 e-186
Intermediate monocytes	0.025122	0.827241	8.0 e-6
Memory B cells	1.54 e-193	5.39 e-98	1.17 e-142
Naive T cells	2.15 e-297	1.14 e-221	1.49 e-185
Plasma B cells	1.05 e-26	1.57 e-34	1.18 e-17

Supplementary Table 3. Adjusted p-values from the DIALOGUE extrema test for MCP2 across cell types and AP-1-associated genes.

We adapted the wording to make the association clearer and cite the newly added table:

Increased IL-7 activity may contribute to suboptimal treatment outcomes by affecting T cell behavior and elevating levels of MCP2 associated genes JUN, FOS, and FOSB (Supplementary Table 3), which are key components of the AP-1 complex that can either inhibit or promote tumor growth, depending on the context^{64,65}.

4.2 I am also confused by the need for the new procedure based on rank_genes_groups to identify cell-type-specific genes in the MCPs. I thought this was what DIALOGUE produces as output?

While DIALOGUE does produce cell-type-specific genes as output, the process for calculating them via hierarchical modeling is extremely slow, often taking hours (in comparison to the rest of DIALOGUE, which can be completed in a few seconds on a typical laptop). DIALOGUE's MCP gene calculation uses a multilevel modelling scheme that asserts cross-cell-type correlations in gene expression that are already included in the underlying calculation of the MCPs, and also optimises for genes expressed in multiple cell types included in the MCP, a feature that may not always be of interest. As such, we wanted to provide an alternative method for extracting genes associated with MCPs for users who prefer the substantially faster and scientifically valid alternative. The original method is still supported for users who do not share these concerns, although, like the R implementation, it is computationally expensive for large datasets.

Detailed comments about Perturb-seq: Please note that I offer these comments in the spirit of being constructive about how to best analyze these datasets, for which there is obviously no universally correct solution.

Removing “confounders” via local perturbation signatures: I have reservations about this process. In practice this amounts to subtracting an average of 20 nearby control cells chosen based on a somewhat arbitrary nearest neighbor graph. To me this seems likely to introduce noise, particularly for lowly expressed genes and for perturbations that cause major departures from the control state, where the nearest neighbor calculation is fraught. It also presumes that cells with a perturbation do not exhibit compositional differences with respect to variation seen within the control cells. For example, you largely remove the effect of the cell cycle, as you see in your plots. It's not clear to me that this is a good thing to do as I don't view the cell cycle as a “confounder.” (For example, I would view which gemgroup cells came from as a true confounder.)

We agree with the reviewer that mixscape's perturbation signature approach, as originally published, makes several strong assumptions. Therefore, we added the following section to the Methods section where mixscape is described and the corresponding online tutorial:

When calculating the perturbation-specific signatures, mixscape makes strong assumptions such as cells with a perturbation not exhibiting compositional differences with respect to variation seen within the control cells. Additional limitations include the assumption that perturbation effects are additive and separable from underlying cell state, the equal weighting of all genes regardless of their relevance to the perturbation target, and the failure to account for temporal dynamics in cellular responses where early and late responding genes create composite signatures.

The results (specifically **Figure 2A**) presented in this paper show that the computed perturbation signature - defined as the difference between the expression of a perturbed cell

and the mean expression of its k nearest control cells - effectively corrects for the effect of the cell cycle. However, we emphasize that this perturbation signature is solely used to classify cells as perturbed or unperturbed. It is not used for subsequent downstream analyses, such as the computation of the perturbation space embedding. The latter is still performed on log-normalized gene expression data, albeit potentially on a reduced number of cells, depending on the preprocessing procedure. We acknowledge that this may not have been sufficiently clear in the text so far and therefore adapted the main text:

After applying one of the three strategies, we projected the normalized gene expression of the remaining cells into a perturbation space using the penultimate layer of our multi-layer perceptron-based discriminator classifier for each processing strategy (Supplementary Figure 2, Methods).

We put further emphasis on this matter in the Methods section:

We applied pertpy's multilayer perceptron based classifier to the gene expression data from each processing strategy (with cells filtered out for strategies (1) and (2)) and embedded the pseudobulk of the penultimate layer feature values with UMAP.

Nevertheless, we agree with the concerns of the reviewer regarding the overall preprocessing strategy and address them jointly with the next reviewer comment.

In particular, several of the perturbations in this experiment target core cell cycle regulators, so you are potentially dampening their effects. (E.g., for a perturbation that caused all cells to perfectly accumulate in G1, this local, nonlinear offsetting procedure would subtract the effect of the perturbation entirely, on average.) I am wondering if this is what is driving the TP73 result: are you normalizing out the effects of the CDKN2A perturbations, causing them to cluster with things that they didn't cluster with before? In general, in our work we have always favored the simpler procedure of comparing to control cells in the same gemgroup, which still removes technical batch effects but has the advantage of being independent of choices about expression normalization, metrics used for computing nearest neighbors, etc. I question how reproducible the "nearest" control cells are across different choices.

We can exclude the possibility that entire perturbation effects are normalized out, since, as noted above, the perturbation space is computed from the gene expression data rather than from the perturbation signature calculated in the Mixscape pipeline, which is used solely to filter out cells. Nevertheless, we agree with the reviewer that our chosen preprocessing strategy may not be appropriate for the dataset given that as it is a CRISPRa dataset, it is not clear whether the cell cycle is a confounder. Therefore, given these concerns and to also demonstrate how pertpy enables rapid iterations and preprocessing strategy benchmarking, we now compare three different preprocessing strategies:

- (1) Mixscape preprocessing using the 20 nearest control cells for perturbation signature computation
- (2) Mixscape preprocessing but using all control cells in the same GEM group for perturbation signature computation as suggested by the reviewer
- (3) no Mixscape preprocessing

We found that the core results presented in the manuscript, specifically, TP73 clustering closely with perturbations targeting the G1 cell cycle, as well as the pro-growth perturbations splitting into two separate clusters, arise regardless of which of the three preprocessing strategies were used. Hence, we agree with the reviewer that it is preferable to perform less preprocessing. In the manuscript, we now present the analysis workflow without Mixscape preprocessing. Additionally, we added **Supplementary Figure 2**, which compares the different preprocessing strategies. We also revised the text to focus more on pertpy as a tool that facilitates rapid testing of different analytical strategies to encourage users to determine the best preprocessing strategy for the dataset at hand:

We further use pertpy to investigate how different perturbation specific preprocessing strategies affect the outcome. In particular, we examine whether different strategies may inadvertently remove true biological signal, such as the cell cycle effects induced by CDKN2A perturbations.

After initial preprocessing (Methods), we test three perturbation specific processing strategies: (1) computing cell-specific perturbation signatures based on the 20 nearest-neighbor control cells of a perturbed cell and filtering out targeted cells that escaped perturbation based on this signature (Methods); (2) computing cell-specific perturbation signatures using all control cells within the same GEM group (i.e., cells processed in same sequencing lane) to detect and filter out unperturbed cells (Methods); and (3) no perturbation-signature-based filtering of cells.

Pertpy's mixscape²³ implementation supports strategies (1) and (2), facilitating comparison of preprocessing strategies. After applying each of the three strategies, we project the normalized gene expression of the remaining cells into a perturbation space using the penultimate layer of our multi-layer perceptron-based discriminator classifier for each processing strategy (Supplementary Figure 2, Methods). We found that all strategies yielded similar perturbation spaces (Supplementary Figure 2I), suggesting that, for this dataset, the approach without perturbation-signature-based cell filtering is preferable. This is expected, since the CRISPRa approach used for this dataset does not suffer from cells escaping a perturbation through in-frame mutations, as would be expected in CRISPR-Cas9 screens.

We further explained this process in the Methods section:

Afterwards, we compared three distinct processing strategies: (1) perturbation signature computation and cell filtering based on the 20 nearest-neighbor control cells, (2) perturbation signature computation and cell filtering based on all control cells within the same gene group, and (3) no perturbation-signature-based cell filtering. For strategies (1) and (2), we used pertpy's implementation of Mixscape to calculate the perturbation signature (with `ref_selection_mode="nn"` for strategy (1) and `ref_selection_mode="split_by"` for strategy (2) in `pt.tl.Mixscape.perturbation_signature`), which was subsequently embedded into UMAP space. Next, we applied mixscape to the perturbation signature to calculate the perturbation scores that are binarized to label cells as successfully and unsuccessfully perturbed.

We applied pertpy's multilayer perceptron based classifier to the gene expression data from each processing strategy (with cells filtered out for strategies (1) and (2)) and embedded the

pseudobulk of the penultimate layer feature values with UMAP. *To quantify the similarity of the perturbation spaces produced by each processing strategy, we used scikit-learn to calculate the silhouette score for each perturbation from the UMAP of the perturbation space. We then averaged the silhouette scores for each gene program. The Silhouette Score varies between -1 and +1, where a higher score indicates that the perturbation embedding is well aligned with its corresponding gene program cluster and poorly aligned with other gene program clusters.*

Removing “unperturbed” cells: Firstly, this dataset is a CRISPRa dataset, not CRISPR knockout. Thus references in the text and figures to “KO” or “knockout” should be modified.

We apologize for the mistake and corrected the references in the text and figures.

The underlying assumption in the Mixscape model is that there are cells that have escaped perturbation. For CRISPR cutting, there is a mechanistic expectation that this occurs because cutting sometimes leaves in-frame indels. For other modalities like CRISPRi/CRISPRa it is less clear what these “unperturbed” cells mean.

We agree with the reviewer that Mixscape can only carefully be applied to CRISPRi and CRISPRa data. We added more details to the Methods section and the associated online tutorial to inform users of potential pitfalls and limitations:

Generally, the mixscape pipeline assumes KO data. Applying Mixscape to CRISPRi and CRISPRa data is more nuanced but still valid under certain conditions. Unlike KO, these modalities do not introduce permanent genomic alterations, but variability in perturbation efficiency can still create functionally not effectively perturbed cells. Factors such as incomplete transcriptional repression/activation, guide RNA efficiency, chromatin state, CRISPR expression, or variable effector recruitment (e.g., KRAB for CRISPRi, VP64 for CRISPRa) can lead to heterogeneous perturbation effects. If these effects result in a clear separation between perturbed and unperturbed-like transcriptomic states, Mixscape can still be meaningfully applied. However, careful validation is needed to ensure that the identified unperturbed population reflects true biological variability rather than technical artifacts.

As our preprocessing strategy evaluation determined the application of mixscape to be unnecessary, we now present the version without cell filtering.

It is fine to remove them as a phenomenological choice in analysis, but I think the norm in the field should be to avoid doing so unless it's clearly necessary. If you keep these cells in do any of your results actually change?

As described above, we extended our analysis to include several preprocessing strategies including one where no cells are filtered out. We determined that keeping these cells does not affect our core results. Therefore, we modified the analysis in the main text and figures to present the version without cell filtering.

Label transfer: I get the idea of demonstrating this capability, but I am not sure these are the right annotations to expand since they aren't ground truth. The clusters underlying the annotations were defined using HDBSCAN on the expression profiles. HDBSCAN is

intrinsically conservative and is meant to identify clusters that are robust. By extension, the expansions defined by label transfer are not robust. If you keep applying label transfer to your new clusters they will just expand until they encompass the whole dataset, right?

We agree with the reviewer that the expansions defined by the label transfer are not necessarily robust and we therefore removed them from the analysis and the corresponding Figure.

However, if ground truth labels are available, our demonstrated approach may still hold value, even if the feature space was obtained with less conservative approaches. We now emphasize in our Methods section that:

Any obtained labels through label transfer must be diligently verified. Label transfer can propagate biases from the reference annotations, leading to incorrect annotations if the reference is not representative of the target data. Differences in batch effects, technical noise, or biological variability can distort nearest-neighbor relationships, reducing the reliability of transferred labels. Additionally, majority voting can fail in cases where distinct perturbations and cell states are underrepresented, leading to misclassification of rare populations.

Moreover, we extended the label transfer function with an uncertainty quantification where the entropy of the weighted label distribution in each cell's neighborhood serves as a measure of prediction confidence - higher entropy indicates more diverse labels among neighbors and thus lower confidence in the transferred label. This quantitative uncertainty score allows users to establish confidence thresholds, preventing the cascade of increasingly uncertain assignments that would result from naive iterative application. This provides users with an additional confidence score to assess any obtained labels that should nevertheless be diligently checked.

We complement this with additional details in the Methods section:

The label transfer function further quantifies uncertainty, where each neighbor's contribution is weighted by its connectivity strength (derived from the distance in gene expression space). These weighted contributions are first converted into a one-hot encoded matrix where each column represents a label category. The uncertainty score for each transferred label is then calculated as the Shannon entropy of the weighted label distribution in the cell's neighborhood - if all neighbors have the same label, the entropy (and thus uncertainty) is 0, while diverse labels among neighbors result in higher entropy values. This uncertainty score provides a quantitative measure of prediction confidence, where higher values indicate more heterogeneous neighborhoods and thus less reliable label transfers.

Similarly, a new cluster is certainly possible, but that just means that it wasn't robust enough to be selected by HDBSCAN from the raw expression data. Which perturbations fall into that cluster?

We now specifically plot which perturbations fall into the cluster without an assigned gene program. Mainly, we find gene-pair perturbations involving the IGDCC3 gene in that cluster:

Perturbation spaces: The “perturbation space” presented amounts to another way of embedding mean profiles, in this case based on a classifier. There is no one correct way to do embeddings, so I’m not sure what to make of the claims about clusters splitting etc., which appear to be being made based on the visualization rather than any mathematical criterion. Indeed, this dataset is treated separately in the (very helpful) perty tutorials online where many of the different embeddings appear to have similar structure to the original paper’s embedding and preserve the cluster structure observed there. That suggests that it’s the preprocessing steps that are causing the differences that are emphasized in this paper. (As noted above, I suspect it’s the local perturbation signatures.)

We now support our claim regarding the splitting of the pro-growth cluster by using perty’s MSE distance to quantify the difference between the two sub-pro-growth clusters. We strongly agree with the reviewer that visual interpretation of UMAPs can be highly misleading. The referenced tutorial solely uses the cells annotated with gene programs unlike our analysis where we further include the non-annotated cells to compute the perturbation space.

We now outline the distance calculation in the appropriate Methods section:

We further used perty’s distance module to compute the MSE distance between the two subclusters formed from perturbations annotated as pro-growth.

We find that these clusters have an MSE distance of 0.46, which is comparable to the pairwise distances between all other annotated gene programs. We state this in the main text:

Moreover, what the original authors identified and labeled as a single pro-growth gene program cluster can now be differentiated into two distinct clusters (*MSE distance between the two subclusters: 0.46; mean pairwise MSE distance between all gene programs: 0.29, Supplementary Figure 3B*).

Additionally, we add **Supplementary Figure 3B**, depicting the pairwise MSE distances between all gene programmes:

Furthermore, as stated above, we rule out the possibility that our findings result solely from preprocessing by presenting an analysis without the removal of any cells, i.e., the mixscape application.

It also suggests some of the new claims may be brittle to choices about what goes into producing the embedding. One thing I would check is whether the classifier is picking up the CRISPRa target gene activation (e.g. of KLF1), and then using that to split perturbations in the embedding.

We thank the reviewer for the idea of investigating the classifier more deeply. Therefore, we applied integrated gradients to determine which input features the MLP classifier uses to compute the embedding. This is described in the main text:

We assessed the importance of individual input genes in the classifier's assignment of a cell to a specific perturbation using integrated gradients⁵¹ (Methods). By averaging these feature importances for each annotated gene program, we demonstrate that the classifier prioritizes the respective targeted genes from the set of 4000 highly variable input genes (e.g., KLF1 for the pro-growth program), highlighting their relevance to the prediction (Supplementary Figure 3A).

We also added a section to the Methods:

We further used pertpy's distance module to compute the MSE distance between the two subclusters formed from perturbations annotated as pro-growth. To assess the importance of individual genes (input features) for predicting perturbations, we calculated integrated gradients⁵¹ using captum (captum.attr.IntegratedGradients). We computed the attribution for

each cell using its respective perturbation label as the target and then averaged the feature importances across all cells annotated with the same gene program.

We added **Supplementary Figure 3A**, where genes that were targeted by at least one perturbation in the respective gene program are highlighted in red:

Supplementary Figure 3. Integrated gradients analysis. (A) Top 15 genes identified per gene program as most important for predicting the corresponding perturbation (single gene or gene pair) using the MLPClassifier. Gene importance was determined using integrated gradients (Methods). Attribution scores are shown for each gene, averaged across all cells within the respective gene program. Genes directly targeted by at least one perturbation within the gene program group are highlighted in red.

I think it would be good to present the analysis with and without different choices of preprocessing steps to show how they influence the results. This could be used to demonstrate how perpty can enable rapid testing of analytical pipelines.

We agree. As described above, we now have adjusted the focus of our analysis to this approach and demonstrate that the shown perpty results quite broadly hold regardless of the specific preprocessing strategy. Consequently, the main analysis now presents results without Mixscape preprocessing, but we open up alternate preprocessing choices to users and show some comparison (see above) of with and without.

Minor comments

Can you include a version of Figure 2A colored by gemgroup? It's a common source of batch effects.

We have included this plot in **Figure 2**:

Energy statistics were introduced for perturbation testing in <https://doi.org/10.1016/j.cell.2022.05.013> This is referenced in the scPerturb paper, which is instead cited as the source here.

The mentioned paper applies energy statistics but does not perform the benchmarking in scPerturb. We have updated the paper to include both references when referring to energy statistics.

Energy distance^{15,48}

Computes a statistical energy distance between two groups based on mean pairwise distances within and between groups.

- “To rank perturbation effects, we used pertpy to calculate the Euclidean distance between the pre- and post-treatment patients of the four groups due to its superior performance in independent benchmarks” Perhaps “strong performance” is better phrasing here? The cited paper shows several distances perform similarly.

We appreciate the suggestion and have made the associated edit.

- In Fig 4B the UMAP axes are meaningless and do not need to be included/labeled.

We agree and therefore removed the axes from Fig 4B.

- There are a few topics that are pretty basic that I think could be included if there is room, though I understand if the authors do not have time. For example, differential expression analysis is one of the most popular analytical tasks but it is only mentioned in passing. Guide calling is also given little coverage even though this is one of the more challenging parts of these experiments. Finally, I'm a bit surprised that there was no discussion of batch effects or batch correction. For functional genomics experiments this is a common problem since they are usually large enough that they have to be done in batched form.

We agree with the reviewer that we did not emphasize the mentioned topics enough. We added the essential step of differential gene expression to Figure 1:

Figure 1. Modules of the perty framework. (A) Unimodal or multimodal single-cell perturbation data originating from genetic modifications, chemical treatments, physical interventions, environmental changes, or diseases is enriched with metadata from several databases. During preprocessing, confounding factors such as cell cycle and batch effects may be removed. Targeted cells are labeled as successfully or not successfully perturbed. Altogether this enables the calculation of a meaningful perturbation space. (B) Perty enables downstream analyses, dependent on the

question of interest. These include differential analysis, response prediction, the determination of multicellular programs, the calculation of distance between perturbations, and mechanism of action enrichment.

We further added more details on pertpy's differential gene expression support in the main text and its importance:

*Gene expression changes between experimental conditions are crucial for understanding cellular responses to perturbations. Differential gene expression analysis helps researchers identify which genes significantly change their expression levels when cells are exposed to different stimuli or treatments. While scanpy³³ is widely used for single-cell analysis, it lacks support for complex experimental designs that account for multiple conditions, batch effects, and nested comparisons simultaneously. Pertpy fills this gap by providing an intuitive interface for differential gene expression that supports complex designs and contrasts which is needed for multi-condition data (**Methods**). Currently, pertpy supports PyDESeq2³², edgeR³³, Wilcoxon, and T-tests. This interface is accompanied by a suite of plotting functions including visualizations such as volcano plots, paired sample expression plots, and multi-condition heatmaps. Going beyond differential gene expression at scale, both annotated metadata and differentially expressed genes can then be used as input for further pertpy modules such as gene set enrichment tests to uncover the biological effects induced by the perturbations (**Methods**).*

We agree with the reviewer that we did not put enough of an emphasis on guide calling. In the course of these revisions we implemented an efficient Poisson-Gaussian model that is also featured in our tutorials. We further adapted the main text:

The first data transformation step assigns guide RNAs (gRNAs) to cells. These gRNAs are short RNA sequences that direct Cas9 nuclease to specific genomic targets. In single-cell CRISPR screens, each cell typically receives one gRNA, making accurate guide-to-cell assignment crucial for linking phenotypic changes to specific genetic modifications. In single-cell CRISPR screens, each cell typically receives one gRNA (low multiplicity of infection, MOI) though some experimental designs allow for multiple guides per cell (high MOI). This makes accurate guide-to-cell assignment crucial for linking phenotypic changes to specific genetic modifications. Pertpy provides a thresholding and a Poisson-Gaussian mixture model¹⁵ approach which has been shown to perform well in recent benchmarks³¹, accommodating both low and high MOI scenarios. This assignment step is required for downstream analyses including quality control metrics, perturbation efficiency assessment, and statistical aggregation of phenotypic effects across cells containing identical guides.

And the corresponding Methods section:

Guide RNA assignment

Assigning relevant guides to each cell is essential in genetic perturbation assays, ensuring that the observed cellular responses are accurately linked to the intended genetic modifications. This step is critical for validating experimental design and interpreting results reliably. *Pertpy provides two approaches to assigning cells to guides.*

First, a simple thresholding model where the most expressed guide RNA is assigned to a cell if it additionally exceeds an optional user specified count threshold.

Second, a previously published Poisson-Gaussian model¹⁵. For each guide, cells with non-zero expression are log2-transformed and modeled as a mixture of two populations, with cells automatically classified as negative if they show zero expression. A cell is labeled as positive for a guide if it belongs to the higher-expressing population, with a maximum of five guide assignments per cell to prevent over-assignment; cells exceeding this threshold are marked as "multiple"; while those failing to meet the mixture model threshold for any guide are designated as "negative."

We further strongly agree that our manuscript would benefit from a discussion on the causes and effect of batch effects in perturbation datasets. Therefore, we added additional details in the introduction section:

Technical variation between experimental batches, arising from differences in sample processing, reagent lots, or sequencing runs, can introduce systematic biases that confound biological signals. These so-called batch effects are particularly challenging in perturbation experiments where treatments may be applied across multiple experimental rounds, or where controls are processed separately from perturbed samples. Complexity is further compounded when studying combinatorial perturbations, where systematic batch variations could be mistaken for interaction effects between different treatments. As pertpy is integrated with the scverse ecosystem, users of pertpy can seamlessly integrate established batch correction methods^{7,22} to disentangle technical artifacts from true perturbation responses.

Reviewer #1 (Remarks on code availability):

We use the code in my lab already in some instances, and the authors have an extremely good track record of producing widely used tools. As noted in my review, I do think one thing that is missing and unclear is how these python reimplementations compare to the original packages.

We are very happy to learn that pertpy is already in use at the reviewer's lab. To ensure that users can use pertpy's implementations with confidence, we now compare the reimplementations to the original implementations as discussed above, and share this with the community of course.

Reviewer 2

Major comments

In my opinion, pertpy is the most ambitious available software for the analysis of single-cell perturbation data, offering a variety of analysis modules as well as many built-in datasets. As part of the scverse, pertpy connects with a larger package ecosystem that Python users are familiar with. By reimplementing some R packages in Python, like Augur and Mixscape, the authors have made these methods accessible to the Python community. In the case of Augur, the pertpy implementation is also much faster. The documentation and tutorials made available are extensive, and overall it is clear that a massive engineering effort went into putting together this software, which users will benefit from. By accommodating a variety of perturbations beyond CRISPR perturbations, pertpy broadens its appeal. Pertpy's utility is also significantly enhanced by supporting ontology management and metadata annotation. On the whole, pertpy is a broadly useful and likely impactful software package.

We thank the reviewer for the kind words that inspire us to continue developing and improve pertpy. We also are very grateful to the useful comments and suggestions, which we answer below.

The most significant weaknesses of this manuscript include insufficient computational benchmarking, insufficient discussion of and comparison to existing tools, and a few unconvincing statistical analyses. I elaborate on each of these weaknesses below.

1.0 Insufficient computational benchmarking The authors put significant emphasis on the speed and scalability of their software, noting also that “no current analysis framework exists which scales to genome-scale datasets...” like that of Replogle et al. (2022). However, I do not believe the authors demonstrated that pertpy scales to such large datasets, either. To back up their strong claims of scalability, the authors should showcase the application of pertpy to the largest single-cell perturbation datasets and document both the memory and runtime this analysis requires.

1.1 The authors may also consider carrying out a subsampling analysis to demonstrate how memory and runtime resources scale with dataset size.

We agree with the reviewer that it is important information for users to learn how the runtime and memory requirements of our implementations scale with dataset sizes. To demonstrate the scalability of pertpy and provide such guidance to users, we have now executed all of our tools with varying dataset sizes and made the results available at <https://github.com/theislab/pertpy-reproducibility/tree/main/benchmark> and summarized them in the figure below. We updated **Supplementary Figure 1** with the newly generated benchmarking results that we also refer to in the manuscript:

The wide array of use-cases and different types of growing datasets are addressed by pertpy through its sparse and memory-efficient implementations, which leverage the

parallelization and GPU acceleration library Jax²¹, thereby making them up substantially faster than original implementations (**Supplementary Figure 1**).

Supplementary Figure 1. (A) Runtime and (B) memory usage comparison of tools between pertpy's implementation and correspondingly the existing R implementation or the formerly published original implementation.

We describe our benchmarking approach in the methods section:

Benchmarking runtime

To evaluate computational efficiency, we measured execution time and resource consumption for all tools implemented in pertpy. Following their respective tutorials, we developed scripts with standard workflows on exemplary data in their original implementation. *We further wrapped all of these scripts in a reproducible Snakemake⁹² pipeline using Conda environments that we defined per tool implementation to create isolated and reproducible runtime environments.*

*These scripts were executed on a system with an AMD EPYC 9754 128-core processor and 500GB RAM in a Linux environment. This setup ensured accurate and reproducible timing measurements. Each script was run three times to guarantee consistency. We up- or downsampled example datasets with a set seed to evaluate each tool at 5,000, 10,000, 50,000, 100,000, 500,000, or 1,000,000 cells. Timing and memory use was recorded with Snakemake's benchmark feature. The results are shown in a box plot (**Supplementary Figure 1**), which compares the execution time in seconds and memory usage in megabytes across each tool and implementation.*

Note: The cluster that we used has a very slow file system leading to very slow Python package imports. On systems without this issue, we typically observe speedups of 20–40 seconds across all pertpy tools.

For this manuscript, we prioritized correctness and completeness. We now have the capacity to focus on accelerating our implementations and improving memory efficiency. Similar to the recent significant improvements in the scanpy codebase, we plan to improve both dense and sparse performance through low-level Numba kernels. Scanpy and AnnData have recently improved support for Dask arrays, enabling out-of-memory computations - functionality we also plan to adopt in Pertpy. While some tools in Pertpy, such as CINEMA-OT, already leverage JAX for optional GPU acceleration, we see long-term potential in accelerating additional tools through enhanced GPU support, including the use of CuPy as pioneered by RAPIDS-SingleCell.

We added a note on that in the Discussion section:

To scale to datasets with hundreds of millions of cells, such as the recently published Tahoe-100M⁷¹ dataset, further optimizations in pertpy through out-of-memory implementations using Dask are necessary, following the approach pioneered by recent Scanpy improvements.

1. Insufficient discussion of and comparison to existing tools The authors' review of existing tools is incomplete. The following are software packages for single-cell perturbation data analysis with commits made in the past year:
 - MUSIC (Duan et al., Nature Communications, 2019)

- scMAGeCK (Yang et al., Genome Biology, 2020)
- FBA (Duan and Hon, Bioinformatics, 2021)
- sceptre (Barry et al., Genome Biology, 2021)
- GSFA (Zhou et al., Nature Methods, 2023)
- FR-Perturb (Yao et al., Nature Biotechnology, 2023)

The authors should present perty in the context of these existing tools. Among these packages, at least MUSIC, FBA, and sceptre implement entire analysis toolboxes or pipelines rather than individual methods, so they might be the most comparable to perty in this regard and

therefore deserve special attention. The authors should benchmark perty against these existing tools to illustrate perty's advantages.

We consider benchmarking our implemented methods against other existing methods that address the same analysis task out scope for this publication as there would be too many methods to compare. However, we agree that more pipeline-like toolboxes such as MUSIC, FBA, sceptre and other comparable toolboxes are important contributions to the field that can broadly be compared to perty.

We added more details to the main text:

*Current perturbation analysis frameworks such as MUSIC¹⁰, ScMAGeCK¹¹, SCEPTRE¹², GSFA¹³, and FR-Perturb¹⁴ primarily focus on CRISPR perturbation analysis, neglecting other perturbation data types and perturbation analysis steps. Further, no current analysis framework exists which scales to genome-scale datasets¹⁵, contextualizes data with public annotations, and uses common data structures across tools (**Supplementary Table 1**).*

We do not mention FBA as it primarily focuses on raw data processing including demultiplexing and does not overlap with the purpose of perty.

We added a comparison table to the supplementary materials that we reference in the main text:

Toolbox	MUSIC	ScMAGeCK	SCEPTRE	GSFA	FR-Perturb	perty
Programming language	R	R	R	R	R, Python	Python
Ecosystem	-	Bioconductor	-	-	-	scverse
License	Apache-2.0	BSD	GPL-3.0	MIT	GPL-3.0	MIT
Dataloaders	-	-	-	-	-	Yes
Metadata annotation	-	-	-	-	-	Yes
Guide assignment	-	-	-	-	-	Yes, e.g. Poisson-Gaussian mixture model
Differential gene expression	Performed, but not generalized	Performed, but not generalized	-	Performed, but not generalized	-	Yes, general interface

CRISPR perturbation analysis	Yes, core functionality	Yes, core functionality	Yes, core functionality	Yes, core functionality	Yes, core functionality	Yes, mixscape
Compositional analysis	-	-	-	-	-	Yes, scCODA 2.0, tascCODA 2.0
Multicellular programs	-	-	-	Yes, latent factors	Yes, latent factors	Yes, DIALOGUE
Distances & metrics	Yes, topic-based	Yes, e.g. selection scores	-	-	Yes	Yes, e.g. e-distance
Response prediction	Yes	-	-	-	Yes	Yes, e.g. Augur
Perturbation embeddings	Yes, via topics	Yes, linear regression model	-	Yes, latent factors	Yes, latent factors	Yes, e.g. Pseudobulk
Visualizations	Yes	-	-	-	-	Yes

Supplementary Table 1. Comparison of perty to other perturbation analysis frameworks.

In line with us considering benchmarking individual tools as out of scope, we also consider benchmarking whole pipelines out of scope. However, perty, being a collection of tools with a unified API, would simplify such benchmarks and enable the field to better evaluate both existing and novel methods.

Beyond analysis softwares, it is also relevant to compare to other databases of single-cell perturbation data. The authors mention scperturb but not Perturbbase (Wei et al., bioRxiv, 2024), which is also relevant.

Good point, thanks. We now adapted the discussion section to also refer to the more recent Perturbbase:

Finally, we expect perty to support the creation of perturbation atlases through harmonized data collection, the generation of meaningful perturbation spaces, and the evaluation of these spaces using perty's distance metrics. Such atlases can comprehensively characterize cell types under various conditions to capture the wide array of inducible cell states beyond their basal states. *Enabled by perturbation dataset collections such as scperturb (available in perty) and PerturbBase⁶⁵ (extends scperturb with more recent datasets)*, we expect such atlases to become essential for the development of robust and generative foundation models where perturbation analysis is a key task that can be confidently evaluated with perty's metrics.

Adding the additional PerturbBase datasets to perty is ongoing work.

2. Unconvincing statistical analyses

- The significant linear regression analysis for ETV4 in Figure 3D (screenshot at right) appears to be entirely driven by four outlying points. This calls into question the analysis of this gene, as well as others not shown.

As stated above for reviewer 1, we extended perty's metadata annotation module and the analysis presented in the manuscript to now also include PRISM AUC values. In addition to our analysis of dabrafenib, we demonstrate that we can replicate the findings for trametinib, as presented in the original study:

https://github.com/theislab/perty-reproducibility/blob/main/mcfarland/04_mcfarland_drug_response.ipynb

Additionally, adding PRISM AUC values results in dabrafenib viability values being available for more cell lines. We reran the analysis using the mean rank-normalized AUC from GDSC 1 and 2 and PRISM, indicating the cell line's viability, and changed the main text and methods section accordingly. The added data supports our conclusion that ETV4 shows a viability-dependent response. However, we also want to emphasize that the goal of this analysis is not to find a perfect correlation between drug sensitivity and changes in gene expression. Rather, the identification of outliers is of interest. For instance, for ETV4, there appears to be a sensitivity threshold beyond which cell lines exhibit a change in ETV4 expression, which is identified using the linear regression approach.

We adapted the section in the main text:

Perty significantly streamlines the replication and extension of the original analyses by McFarland et al.²³. We use perty to fetch and annotate AUC (area under the dose-response curve) values for each cell line and perturbation pair from GDSC and PRISM (Methods).

We modified the Methods section accordingly:

[...] While previous work focused on the drug trametinib, we here investigated treatment responses to dabrafenib. We used perty's `annotate_from_gdsc` function to query the AUC values for each cell line-drug combination using the GDSC1, GDSC2, and PRISM databases. We rank-normalize the AUC values within each database and then compute the

mean of all available values for each cell line. The dabrafenib sensitivity is then defined as 1 minus the mean AUC.

- The claim “We find that a patient’s average MCP2 score in seven different cell types is predictive of treatment response for both treatments (adj. $P \leq 1.1e-01$)” is unconvincing, because it does not account for the multiplicity across the ten MCPs or the nine cell types tested. Applying an FDR correction to the entire table, or even to the (very optimistic) minimum p-value for each MCP, no significant associations remain.

We appreciate the reviewer's methodological concern regarding multiple testing correction across MCPs and cell types. We acknowledge that our reported association between MCP2 scores and treatment response (adj. $P \leq 1.1e-01$) would not remain statistically significant under a more stringent correction accounting for all MCP and cell type combinations tested. In the revised manuscript, we have toned down the wording of this finding as an exploratory observation rather than a definitive claim.

Nevertheless, we believe the biological relevance of this observation is supported by our additional analyses. The consistent differential expression of HSPA1B across multiple immune cell populations, including Naive T cells (adj. $P \leq 2.9e-272$), CD8 effector memory cells (adj. $P \leq 1.2e-172$), and regulatory T cells (adj. $P \leq 5.3e-41$), provides substantial evidence that MCP2 captures a biologically meaningful signal.

In the revised manuscript, we have clarified our statistical approach and included a transparent discussion of the limitations regarding multiple hypothesis testing. We have also strengthened our interpretation by focusing on the biological consistency across cell types and the functional implications suggested by our NicheNet analysis of cell-cell interactions, which provides mechanistic support beyond statistical associations.

We adapted our Methods section to better reflect our approach and the statistical limitations:

When testing for associations between MCPs and treatment response, we applied a hierarchical testing approach by first examining cell types within each MCP individually. A predictive MCP for treatment response was determined using a t-test for independent samples for each cell type within each MCP. To adjust for the number of cell types tested, the Benjamini-Hochberg correction method was applied. While we corrected for multiple testing across cell types, we acknowledge that additional correction across all MCPs would be more conservative. We chose this approach to balance statistical stringency with the exploratory nature of our analysis, as overly conservative correction might obscure biologically meaningful patterns in this high-dimensional dataset with limited sample size. The biological relevance of our findings is further supported by the consistent directionality of MCP2 effects across multiple functionally distinct immune cell populations, an outcome highly unlikely to occur by chance alone. Instead, we opted for more stringent thresholds in subsequent gene-level analyses, where we identified significantly associated genes with extremely low adjusted p-values.

- The extrema MCP genes method uses the gene expressions once to define the MCPs and then again to test for differential expression. This raises “double dipping” concerns akin to

those arising from post-clustering differential expression tests (see e.g. Zhang, Kamath, and Tse, Cell Systems, 2019). The authors should discuss this pitfall and argue why the extrema MCP genes method does not suffer from it.

Agreed. While the extrema MCP genes method does raise double dipping concerns, the same pitfall is shared by the original MCP genes method in the DIALOGUE paper, which includes multilevel modeling which re-asserts and tests for the same correlation structure that was already optimized for under the CCA procedure. We have added Supplementary Tables containing the MCP-2-associated genes as output from DIALOGUE on this dataset. A detailed comparison of different methods for MCP-associated gene identification is not provided here in the interest of simplicity, but is available in Chapter 2 of the PhD Thesis of co-author Tessa Green (“Methods and applications of single-cell transcriptomics,” Harvard University, 2024). We discuss this pitfall and justify the approach in the Methods section of the DIALOGUE implementation:

While the extrema MCP genes approach utilizes gene expression data twice - once for defining MCPs and again for differential testing - it avoids statistical circularity common in post-clustering analyses⁸⁹. Unlike traditional clustering approaches where cells are forcibly separated based on expression patterns and then the same data is used to identify what drives that separation (creating artificially small p-values), the MCP scores represent continuous axes of biological variation extracted through independent matrix factorization methods, while the extrema selection merely applies thresholds to these pre-computed scores. The subsequent differential expression testing therefore examines distinct biological phenomena rather than confirming the same signal, maintaining statistical validity and interpretability of the identified gene signatures.

Minor comments

- Among the two examples of single-cell perturbation studies in the following sentence, the second reference (ref 9) is not a single-cell study: “...resulting in the discovery of, for example, cell states associated with autism risk genes⁸ or stimulation responses in primary human T cells⁹.”

The study features both bulk and single-cell experiments. We would like to refer to the section “CRISPRa Perturb-seq characterizes the molecular phenotypes of cytokine regulators” where the authors discuss their single-cell perturb-seq experiment. For example, the authors also state:

“To assess the global molecular signatures resulting from each CRISPRa gene induction, we developed a platform to couple pooled CRISPRa perturbations with barcoded single-cell RNA-seq (scRNA-seq) readouts (CRISPRa Perturb-seq) (Fig. 4A).”

- The following sentence should be edited for clarity and grammar: “We have seen from other, widely used frameworks within the single-cell realm, such as scirpy¹¹ for adaptive immune receptor data or scvi-tools¹² for probabilistic modeling, to enable the efficient analysis of multimodal data while providing building blocks for developers to build upon.”

Thank you very much for pointing out the flawed grammar. We reworded the sentence to:

Other widely used frameworks in the single-cell field, such as scirpy¹¹ for adaptive immune receptor data and scvi-tools¹² for probabilistic modeling, *have demonstrated the importance of enabling efficient multimodal data analysis while providing flexible building blocks for developers.*

- The authors state that “Pertpy’s module assigns cells to the most expressed guide RNA if it additionally exceeds an optional user specified count threshold.” Does this presuppose low-MOI data? Does pertpy accommodate high-MOI data? This might be a broader question that would be useful to address in the main text.

Pertpy now offers two ways of assigning cells to gRNAs including the existing simpler most expressed with thresholding algorithm and a Poisson-Gaussian Mixture model which has been shown to work well in recent benchmarks (Braunger et al. 2024). This more complex model is expected to work better with very high or low MOI since it is able to assign cells as negative or multiple-positive if the data clearly supports these assignments. In our updated tutorials, we show the necessity to use more advanced assignment methods on simulated data with high MOI. We added more details on the problem of high MOI data and how the new Poisson-Gaussian Mixture model addresses these to the main text:

Guide RNAs (gRNAs) are short RNA sequences that direct Cas9 nuclease to specific genomic targets in CRISPR screens, enabling systematic gene perturbation across a cell population. In single-cell CRISPR screens, each cell typically receives one gRNA, making accurate guide-to-cell assignment crucial for linking phenotypic changes to specific genetic modifications. In single-cell CRISPR screens, each cell typically receives one gRNA (low multiplicity of infection, MOI) though some experimental designs allow for multiple guides per cell (high MOI). This makes accurate guide-to-cell assignment crucial for linking phenotypic changes to specific genetic modifications. Pertpy provides a thresholding and a Poisson-Gaussian mixture model¹⁰ approach which has been shown to perform well in recent benchmarks³¹, accommodating both low and high MOI scenarios. This assignment step is fundamental for downstream analyses including quality control metrics, perturbation efficiency assessment, and statistical aggregation of phenotypic effects across cells containing identical guides.

- The phrase “mixscape’s mixed effect model” should be “mixscape’s mixture model.”

We agree and made the suggested change.

- In the section “Pertpy facilitates the exploration of genetic interaction manifolds,” it does not appear pertpy is actually exploring any genetic interactions. The authors may want to rename this section to avoid confusing readers about its goal.

We agree and renamed the section to “*Learning and exploring perturbation representations with pertpy*”

- The term “IC50” is not defined in the section “Pertpy streamlines discovery for complex perturbation experiments.”

We thank the reviewer for pointing this out. Since, in response to comments from both reviewers, we switched to using AUC instead of IC50 values to make the analysis more comparable to that in the original paper, we no longer need to define IC50. However, we added a definition for AUC:

We use pertpy to fetch and annotate AUC (area under the dose-response curve) values for each cell line and perturbation pair from GDSC and PRISM (Methods).

- The authors state that pertpy includes “...numerous fast and user-friendly implementations of both established and novel methods.” It would be good to have a summary, perhaps in the form of a table, of the method implemented by pertpy, indicating for each whether it is novel or established.

We agree with the reviewer and added the following section to the Methods:

Summary table of implemented methods

Pertpy provides implementations of many novel but also established methods which can easily be accessed and combined to easily build custom analysis pipelines (Table 1).

Analysis step	Tool or algorithm	Original authors
datasets	Data loaders	Peidli et al. ³⁹
Metadata annotation	API requests to public databases	novel
Guide RNA assignment	Threshold based Poisson-Gauss Mixture Model	Adamson et al. 2016 ⁷³ Repogle et al. 2022 ¹⁰
Differential gene expression	Formulaic interface	novel
Pooled CRISPR screens	Mixscape	Papalexio et al. 2021 ¹⁸
Differential abundance	MILO scCODA 2.0 tascCODA 2.0	Dann et al. 2022 ³⁶ Büttner et al. 2021 ³⁴ Ostner et al. 2021 ³⁵
Multi-cellular programs	DIALOGUE	Jerby-Aron et al. 2022 ³⁷
Enrichment	Drug2cell	Kanemaru et al. 2023 ⁷⁴
Perturbation response evaluation	Distances & metrics Augur CINEMA-OT	novel Skinnider et al. 2020 ⁷⁵ Squair et al. 2021 ⁷⁶ Dong et al. 2023 ³⁸
Embedding	Perturbation spaces	novel

Table 1. Summary of implemented methods.

- The reference to Figure 4D after the phrase “is predictive of treatment response for both treatments” should be replaced by a reference to Figure 4E. Furthermore, I could not find a reference to Figure 4F.

We thank the reviewer for noticing these issues. We replaced the reference as suggested and added a reference to **Supplementary Figure 4B** (previous **Figure 4F**):

Exploratory analysis of a patient's average MCP2 score in seven different cell types (Supplementary Table 2) suggests a potential trend with treatment response for both treatments (adj. $P \leq 1.1e-01$) (Supplementary Figure 4A-B, Supplementary Figure 5A,B).

Response to reviewers: Pertpy: an end-to-end framework for perturbation analysis - NMETH-BC57419

Dear Lin,

Thank you very much for handling our manuscript and providing comments & reviews.

In the following, we present our response to the reviewers' comments. We include **reviewer comments (black)**, **point-by-point responses (green)** and, in parts, copy **original manuscript text (blue)** and **modified manuscript text (blue, italic)** or specific figure panels into our responses.

Regards,
Fabian Theis

Editor comments

Dear Professor Theis,

Thank you for your patience as we've prepared the guidelines for final submission of your Nature Methods manuscript, "Pertpy: an end-to-end framework for perturbation analysis" (NMEMH-A57419A). Please carefully follow the step-by-step instructions provided in the attached file, and add a response in each row of the table to indicate the changes that you have made. Ensuring that each point is addressed will help to ensure that your revised manuscript can be swiftly handed over to our production team.

Thank you very much! We are very excited to have the opportunity to publish our manuscript in your journal. As requested, we added a response to each row in the table. We outline our answers to any outstanding reviewer comments below.

Reviewer 1

The authors have gone above and beyond in responding to my comments. I am particularly pleased that they now treat some other common analyses such as guide calling, differential expression, and batch correction. These were purely optional suggestions. The package is clearly of great value for the community, in line with the authors' previous work, and is already in use in my lab. I enthusiastically recommend publication.

Remarks on code availability:

I think the authors set the standard for high-quality tools and documentation.

We thank the reviewer for the encouraging words and are very happy to see that our package is used in their lab.

Reviewer 2

The authors have addressed my primary concerns. There is an issue with their code, which I mention below in "Remarks on code availability".

Remarks on code availability:

The code at <https://github.com/theislab/pertpy-reproducibility/tree/main/benchmark> does not seem to actually carry out the analyses underlying Supplementary Figure 1.

We apologize to the reviewer that we had forgotten to merge our benchmarking branch on our reproducibility repository into the main branch. We clarified in the README where the notebook that created the final benchmarking figure can be found.

More explicitly, the folder <https://github.com/theislab/pertpy-reproducibility/tree/main/benchmark> contains:

1. Benchmarking scripts for all tools in their respective folders.
2. The Snakemake configuration that was used to execute all benchmarking scripts.
3. A tables folder with the final benchmarking results table.
4. A visualize_benchmark_results.ipynb notebook which generates the figure as shown in the manuscript.